# An empirical study of implicit regularization in deep offline RL

**Caglar Gulcehre**,* **Srivatsan Srinivasan**\*, **Jakub Sygnowski, Georg Ostrovski,**

**Mehrdad Farajtabar, Matt Hoffman, Razvan Pascanu, Arnaud Doucet**
*DeepMind*

**Reviewed on OpenReview:** *https://openreview.net/forum?id=HFfJWx6OIT*

## Abstract

Deep neural networks are the most commonly used function approximators in offline reinforcement learning. Prior works have shown that neural nets trained with TD-learning and gradient descent can exhibit implicit regularization that can be characterized by underparameterization of these networks. Specifically, the rank of the penultimate feature layer, also called *effective rank*, has been observed to drastically collapse during the training. In turn, this collapse has been argued to reduce the model's ability to further adapt in later stages of learning, leading to the diminished final performance. Such an association between the effective rank and performance makes effective rank compelling for offline RL, primarily for offline policy evaluation. In this work, we conduct a careful empirical study on the relation between effective rank and performance on three offline RL datasets : bsuite, Atari, and DeepMind lab. We observe that a direct association exists only in restricted settings and disappears in the more extensive hyperparameter sweeps. Also, we empirically identify three phases of learning that explain the impact of implicit regularization on the learning dynamics and found that bootstrapping alone is insufficient to explain the collapse of the effective rank. Further, we show that several other factors could confound the relationship between effective rank and performance and conclude that studying this association under simplistic assumptions could be highly misleading.

## 1 Introduction

The use of deep networks as function approximators in reinforcement learning (RL), referred to as Deep Reinforcement Learning (DRL), has become the dominant paradigm in solving complex tasks. Until recently, most DRL literature focused on online-RL paradigm, where agents must interact with the environment to explore and learn. This led to remarkable results on Atari (Mnih et al., 2015), Go (Silver et al., 2017), StarCraft II (Vinyals et al., 2019), Dota 2 (Berner et al., 2019), and robotics (Andrychowicz et al., 2020). Unfortunately, the need to interact with the environment makes these algorithms unsuitable and unsafe for many real-world applications, where any action taken can have serious ethical or harmful consequences or be costly. In contrast, in the offline RL paradigm (Fu et al., 2020; Fujimoto et al., 2018; Gulcehre et al., 2020; Levine et al., 2020), also known as batch RL (Ernst et al., 2005; Lange et al., 2012), agents learn from a fixed dataset previously logged by other (possibly unknown) agents. This ability makes offline RL more applicable to the real world.

Recently, Kumar et al. (2020a) showed that offline RL methods coupled with TD learning losses could suffer from an *effective rank* collapse of the penultimate layer's activations, which renders the network to become under-parameterized. They further demonstrated a significant fraction of Atari games, where the collapse of effective rank collapse corresponded to performance degradation. Subsequently, Kumar et al. (2020a)

---

*Indicates joint first authors.

explained the rank collapse phenomenon by analyzing a TD-learning loss with bootstrapped targets in the kernel and linear regression setups. In these simplified scenarios, bootstrapping leads to self-distillation, causing severe under-parametrization and poor performance, as also observed and analyzed by Mobahi et al. (2020). Nevertheless, Huh et al. (2021) studied the rank of the representations in a supervised learning setting (image classification tasks) and argued that low rank leads to better performance. Thus the low-rank representations could act as an implicit regularizer.

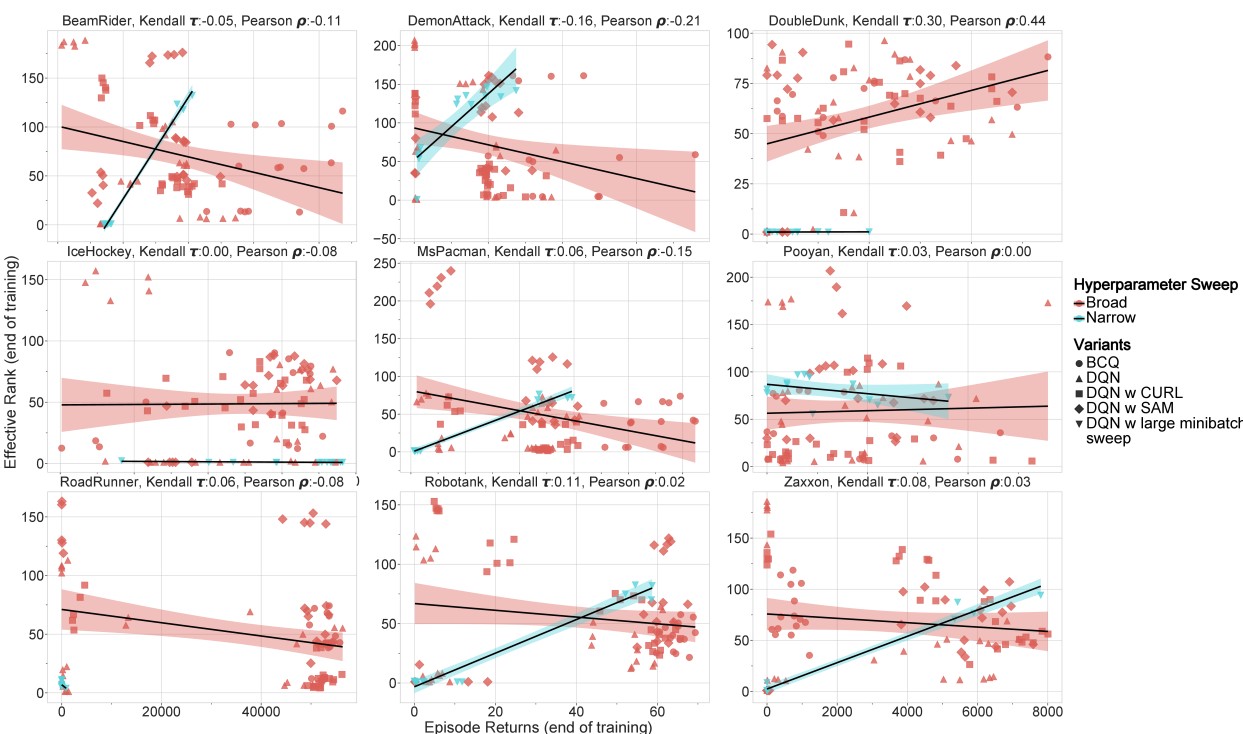

Figure 1: **[Atari] The rank and the performance on broad vs narrow hyperparameter sweep:** Correlation between effective rank and agent's performance towards the end of training in different Atari games. We report the regression lines for the narrow sweep, which covers only a single offline RL algorithm, with a small minibatch size (32) and a learning rate sweep similar to the hyperparameter sweep defined in RL Unplugged (Gulcehre et al., 2020), whereas in the broad setting, we included more data from different models and a larger hyperparameter sweep. In the narrow setup, there is a positive relationship between the effective rank and the agent's performance, but that relationship disappears in the broad data setup and almost reverses.

Typically, in machine learning, we rely on empirical evidence to extrapolate the rules or behaviors of our learned system from the experimental data. Often those extrapolations are done based on a limited number of experiments due to constraints on computation and time. Unfortunately, while extremely useful, these extrapolations might not always generalize well across all settings. While Kumar et al. (2020a) do not concretely propose a causal link between the rank and performance of the system, one might be tempted to extrapolate the results (agents performing poorly when their rank collapsed) to the existence of such a causal link, which herein we refer to as *rank collapse hypothesis*. In this work, we do a careful large-scale empirical analysis of this potential causal link using the offline RL setting (also used by Kumar et al. (2020a)) and also the Tandem RL (Ostrovski et al., 2021) setting. The existence of this causal link would be beneficial for offline RL, as controlling the rank of the model could improve the performance (see the regularization term explored in Kumar et al. (2020a)) or as we investigate here, the effective rank could be used for model selection in settings where offline evaluation proves to be elusive.

> **Key Observation 1:** The effective rank and the performance are correlated in restricted settings, but this correlation disappears when we increase the range of hyperparameters and the types of models that we are evaluating (Figure 1) [a].
>
> ――――――――――
> [a]This is because other factors like hyperparameters and architecture can confound the rank of the penultimate layer; unless the those factors are controlled carefully, the conclusions drawn from the experiments based on the rank can be misleading.

Instead, we show that different factors affect a network's rank without affecting its performance. This finding indicates that unless all of these factors of variations are controlled – many of which we might still be unaware of – the rank alone might be a misleading indicator of performance.

A deep Q network exhibits three phases during training. We show that rank can be used to identify different stages of learning in Q-learning if the other factors are controlled carefully. We believe that our study, similar to others (e.g. Dinh et al., 2017), re-emphasizes the importance of critically judging our understanding of the behavior of neural networks based on simplified mathematical models or empirical evidence from a limited set of experiments.

> **Key Observation 2:** Deep Q-learning approaches goes through three phases of learning: i) simple behaviors, ii) complex behaviors, iii) under-parameterization (Figure 2). These phases can be identified by the effective rank and performance on a given task [a].
>
> ――――――――――
> [a]The first two phases of learning happen during the training of all models we tested. At times, the third phase of the learning could potentially lead to the agent losing its representation capacity and the ensuing poor performance.

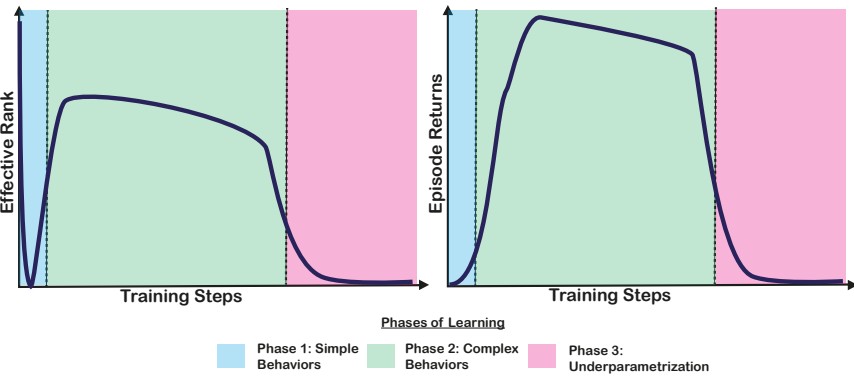

Figure 2: **Lifespan of learning in deep Q-learning:** The plot on the left-hand side illustrates the evolution of the effective rank, and the plot on the right-hand side demonstrates the evolution of the performance during training. In the first phase, the model learns easy-to-learn behaviors that are simplistic by nature and ignore many factors of environmental variations. The effective rank collapses to a minimal value in the first phase since the model does not need a large capacity to learn the simple behaviors. In phase 2, the model learns more complex behaviors that we identify as those that obtain large returns when the policy is evaluated in the environment. Typically, in supervised learning, phase 2 is followed by overfitting. However, in offline RL (specifically the TD-learning approaches that we tried here), we observed that it is often followed by underfitting/under-parameterization in phase 3.

We organize the rest of the paper and our contributions in the order we introduce in the paper as follows:

- Section 2 presents the related work, and Section 3 describes our experimental protocol.

- In Sections 4, 8, 6, 7 and 9, we study the extent of impact of different interventions such as architectures, loss functions and optimization on the causal link between the rank and agent performance. Some interventions in the model, such as introducing an auxiliary loss (e.g. CURL (Laskin et al.,

2020), SAM (Foret et al., 2021) or the activation function) can increase the effective rank but does not necessarily improve the performance. This finding indicates that the rank of the penultimate layer is not enough to explain an agent's performance. We also identify the settings where the rank strongly correlates with the performance, such as DQN with ReLU activation and many learning steps over a fixed dataset.

- In Section 4.1, we show that a deep Q network goes through three stages of learning, and those stages can be identified by using rank if the hyperparameters of the model are controlled carefully.

- Section 5 describes the main outcomes of our investigations. Particularly, we analyze the impact of the interventions described earlier and provide counter-examples that help contradict the *rank collapse hypothesis*, establishing that the link between rank and performance can be affected by several other confounding factors of variation.

- In Section 11, we ablate and compare the robustness of BC and DQN models with respect to random perturbation introduced only when evaluating the agent in the environment. We found out that the offline DQN agent is more robust than the behavior cloning agent which has higher effective rank.

- Section 12 presents the summary of our findings and its implications for offline RL, along with potential future research directions.

## 2 Background

### 2.1 Effective rank and implicit under-regularization

The choice of architecture and optimizer can impose specific *implicit biases* that prefer certain solutions over others. The study of the impact of these implicit biases on the generalization of the neural networks is often referred to as *implicit regularization*. There is a plethora of literature studying different sources of implicit regularization such as *initialization of parameters* (Glorot and Bengio, 2010; Li and Liang, 2018; He et al., 2015), *architecture* (Li et al., 2017; Huang et al., 2020), *stochasticity* (Keskar et al., 2016; Sagun et al., 2017), and *optimization* (Smith et al., 2021; Barrett and Dherin, 2020). The rank of the feature matrix of a neural network as a source of implicit regularization has been an active area of study, specifically in the context of supervised learning (Arora et al., 2019; Huh et al., 2021; Pennington and Worah, 2017; Sanyal et al., 2018; Daneshmand et al., 2020; Martin and Mahoney, 2021). In this work, we study the phenomenon of implicit regularization through the effective rank of the last hidden layer of the network, which we formally define below.

---

**Definition:** Effective Rank

Recently, Kumar et al. (2020a) studied the impact of the effective rank on generalization in the general RL context. With a batch size of $N$ and $D$ units in the feature layer, they proposed the effective rank formulation presented in Equation 1 for a feature matrix $\Phi \in \mathbb{R}^{N \times D}$ where $N \geq D$ that uses a threshold value of $\delta$ with the singular values $\sigma_i(\Phi)$ in descending order i.e. $\sigma_1(\Phi) \geq \sigma_2(\Phi) \geq \cdots$. We provide the specific implementation of the effective rank we used throughout this paper in Appendix A.10.

$$\textbf{effective rank}_\delta(\Phi) = \min_k \left\{ k : \frac{\sum_{i=1}^{k} \sigma_i(\Phi)}{\sum_{j=1}^{D} \sigma_j(\Phi)} \geq 1 - \delta \right\}. \tag{1}$$

---

**Terminology and Assumptions.** Note that the threshold value $\delta$ throughout this work has been fixed to **0.01** similar to Kumar et al. (2020a). Throughout this paper, we use the term *effective rank* to describe the rank of the last hidden layer's features only, unless stated otherwise. This choice is consistent with prior work (Kumar et al., 2020a) as the last layer acts as a representation bottleneck to the output layer.

Kumar et al. (2020a) suggested that the effective rank of the Deep RL models trained with TD-learning objectives collapses because of i) implicit regularization, happening due to repeated iterations over the dataset

with gradient descent; ii) self-distillation effect emerging due to bootstrapping losses. They supported their hypothesis with both theoretical analysis and empirical results. The theoretical analysis provided in their paper assume a simplified setting, with infinitesimally small learning rates, batch gradient descent and linear networks, in line with most theory papers on the topic. We focus our experiments to use the effective rank definition by Kumar et al. (2020a) to make our results more comparable. However, in our preliminary experiments we compared different rank measures and they perfectly correlated to each other on Atari.

## 2.2  Auxiliary losses

Additionally, we ablate the effect of using an auxiliary loss on an agent's performance and rank. Specifically, we chose the "Contrastive Unsupervised Representations for Reinforcement Learning" (CURL) (Laskin et al., 2020) loss that is designed for use in a contrastive self-supervised learning setting within standard RL algorithms:

$$L(s, a, r, \theta) = L_Q(s, a, r, \theta) + \lambda * L_{CURL}(s, \hat{s}, \theta). \tag{2}$$

Here, $L_Q$ refers to standard RL losses and the CURL loss $L_{CURL}(s, \hat{s}, \theta)$ is the contrastive loss between the features of the current observation $s$ and a randomly augmented observation $\hat{s}$ as described in Laskin et al. (2020). Besides this, we also ablate with Sharpness-Aware Minimization (SAM) (Foret et al., 2021), an approach that seeks parameters that have a uniformly low loss in its neighborhood, which leads to flatter local minima. We chose SAM as a means to better understand whether the geometry of loss landscapes help inform the correlation between the effective rank and agent performance in offline RL algorithms. In our experiments, we focus on analyzing mostly the deep Q-learning (DQN) algorithm (Mnih et al., 2015) to simplify the experiments and facilitate deeper investigation into the rank collapse hypothesis.

## 2.3  Deep offline RL

Online RL requires interactions with an environment to learn using random exploration. However, online interactions with an environment can be unsafe and unethical in the real-world (Dulac-Arnold et al., 2019). Offline RL methods do not suffer from this problem because they can leverage offline data to learn policies that enable the application of RL in the real world (Menick et al., 2022; Shi et al., 2021; Konyushkova et al., 2021). Here, we focused on offline RL due to its importance in real-world applications, and the previous works showed that the implicit regularization effect is more pronounced in the offline RL (Kumar et al., 2020a; 2021a).

Some of the early examples of offline RL algorithms are least-squares temporal difference methods (Bradtke and Barto, 1996; Lagoudakis and Parr, 2003) and fitted Q-iteration (Ernst et al., 2005; Riedmiller, 2005). Recently, value-based approaches to offline RL have been quite popular.

**Value-based** approaches typically lower the value estimates for unseen state-action pairs, either through regularization (Kumar et al., 2020b) or uncertainty (Agarwal et al., 2020). One could also include R-BVE (Gulcehre et al., 2021) in this category, although it regularizes the Q function only on the rewarding transitions to prevent learning suboptimal policies. Similar to R-BVE, Mathieu et al. (2021) have also shown that the methods using single-step of policy improvement work well on tasks with very large action space and low state-action coverage. In this paper, due to their simplicity and popularity, we mainly study action value-based methods: offline DQN (Agarwal et al., 2020) and offline R2D2 (Gulcehre et al., 2021). Moreover, we also sparingly use Batched Constrained Deep Q-learning (BCQ) algorithm (Fujimoto et al., 2019), another popular offline RL algorithm that uses the behavior policy to constrain the actions taken by the target network.

Most offline RL approaches we explained here rely on pessimistic value estimates (Jin et al., 2021; Xie et al., 2021; Gulcehre et al., 2021; Kumar et al., 2020b). Mainly because offline RL datasets lack exhaustive exploration, and extrapolating the values to the states and actions not seen in the training set can result in extrapolation errors which can be catastrophic with TD-learning (Fujimoto et al., 2018; Kumar et al., 2019). On the other hand, in online RL, it is a common practice to have inductive biases to keep optimistic value functions to encourage exploration (Machado et al., 2015).

We also experiment with the *tandem RL* setting proposed by Ostrovski et al. (2021), which employs two independently initialized online (active) and offline (passive) networks in a training loop where only the online agent explores and drives the data generation process. Both agents perform identical learning updates on the identical sequence of training batches in the same order. Tandem RL is a form of offline RL. Still, unlike the traditional offline RL setting on fixed datasets, in the *tandem RL*, the behavior policy can change over time, which can make the learning non-stationary. We are interested in this setting because the agent does not necessarily reuse the same data repeatedly, which was pointed in Lyle et al. (2021) as a potential cause for the rank collapse.

## 3 Experimental setup

To test and verify different aspects of the *rank collapse hypothesis* and its potential impact on the agent performance, we ran a large number of experiments on bsuite (Osband et al., 2019), Atari (Bellemare et al., 2013) and DeepMind lab (Beattie et al., 2016) environments. In all these experiments, we use the experimental protocol, datasets and hyperparameters from Gulcehre et al. (2020) unless stated otherwise. We provide the details of architectures and their default hyperparameters in Appendix A.11.

- **bsuite** – We run ablation experiments on bsuite in a fully offline setting with the same offline dataset as the one used in Gulcehre et al. (2021). We use a DQN agent (as a representative TD Learning algorithm) with multi-layer feed-forward networks to represent the value function. bsuite provides us with a small playground environment which lets us test certain hypotheses which are computationally prohibitive in other domains (for e.g.: computing Hessians) with respect to terminal features and an agent's generalization performance.

- **Atari** – To test whether some of our observations are also true with higher-dimensional input features such as images, we run experiments on the Atari dataset. Once again, we use an offline DQN agent as a representative TD-learning algorithm with a convolutional network as a function approximator. On Atari, we conducted large-scale experiments on different configurations:

  (a) **Small-scale experiments:**
      - **DQN-256-2M**: Offline DQN on Atari with minibatch size 256 trained for 2M gradient steps with four different learning rates: $[3e-5, 1e-4, 3e-4, 5e-4]$. We ran these experiments to observe if our observations hold in the default training scenario identified in RL Unplugged (Gulcehre et al., 2020).
      - **DQN-32-100M**: Offline DQN on Atari with minibatch size of 32 trained for 100M gradient steps with three different learning rates: $\left[3 \times 10^{-5}, 1 \times 10^{-4}, 3 \times 10^{-4}\right]$. We ran those experiments to explore the effects of reducing the minibatch size.

  (b) **Large-scale experiments:**
      - **Long run learning rate sweep (DQN-256-20M):** Offline DQN trained for 20M gradients steps with 12 different learning rates evenly spaced in log-space between $10^{-2}$ and $10^{-5}$ trained on minibatches of size 256. The purpose of these experiments is to explore the effect of wide range of learning rates and longer training on the effective rank.
      - **Long run interventions (DQN-interventions):** Offline DQN trained for 20M gradient steps on minibatches of size 256 with 128 different hyperparameter interventions on activation functions, dataset size, auxiliary losses etc. The purpose of these experiments is to understand the impact of such interventions on the effective rank in the course of long training.

- **DeepMind Lab** – While bsuite and Atari present relatively simple fully observable tasks that require no memory, DeepMind lab tasks (Gulcehre et al., 2021) are more complex, partially observable tasks where it is very difficult to obtain good coverage in the dataset even after collecting billions of transitions. We specifically conduct our observational studies on the effective ranks on the **DeepMind Lab** dataset, which has data collected from a well-trained agent on the SeekAvoid level. The dataset is collected by adding different levels of action exploration noise to a well-trained agent in order to get datasets with a larger coverage of the state-action space.

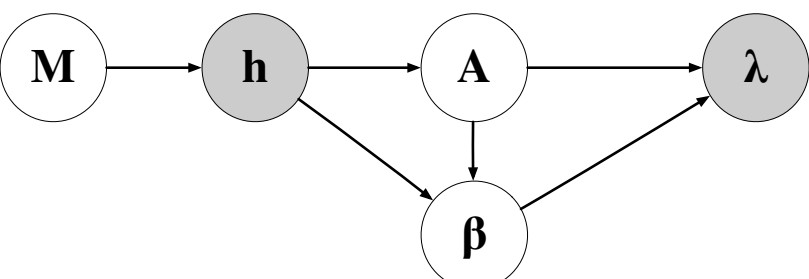

Figure 3: **Structural causal model (SCM) of different factors that we test: M** represents the model selection method that we use to determine the **h** which denotes the observed confounders that are chosen at the beginning of the training, including the task, the model architecture (including depth and number of units), learning rate and the number of gradient steps to train. $\beta$ is the effective rank of the penultimate layer. $\lambda$ is the agent's performance, measured as episodic returns the agent attains after evaluating in the environment. **A** represents the unobserved confounders that change during training but may affect the performance, such as the number of dead units, the parameter norms, and the other underlying factors that can influence learning dynamics. We test the effect of each factor by interventions.

We illustrate the structural causal model (SCM) of the interactions between different factors that we would like to test in Figure 3. To explore the relationship between the rank and the performance, we intervene on **h**, which represents potential exogenous sources of **implicit regularization**, such as architecture, dataset size, and the loss function, including the auxiliary losses. The interventions on **h** will result in a randomized controlled trial (RCT.) **A** represents the unobserved factors that might affect performance denoted by $\lambda$ an the effective rank $\beta$ such as activation norms and number of dead units. It is easy to justify the relationship between M, **h**, **A** and $\beta$. We argue that $\beta$ is also confounded by **A** and **h**. We show the confounding effect of **A** on $\beta$ with our interventions to beta via auxiliary losses or architectural changes that increase the rank but do not affect the performance. We aim to understand the nature of the relationship between these terms and whether SCM in the figure describes what we notice in our empirical exploration.

We overload the term **performance** of the agent to refer to episodic returns attained by the agent when it is evaluated online in the environment. In stochastic environments and datasets with limited coverage, an offline RL algorithm's online evaluation performance and generalization abilities would generally correlate (Gulcehre et al., 2020; Kumar et al., 2021c). The offline RL agents will need to generalize when they are evaluated in the environment due to,

- **Stochasticity** in the initial conditions and transitions of the environment. For example, in the Atari environment, the stochasticity arises from sticky actions and on DeepMind lab, it arises from the randomization of the initial positions of the lemons and apples.

- **Limited coverage:** The coverage of the dataset is often limited. Thus, an agent is very likely to encounter states and actions that it has never seen in the training dataset.

## 4 Effective rank and performance

Based on the results of Kumar et al. (2020a), one might be tempted to extrapolate a positive causal link between the effective rank of the last hidden layer and the agent's performance measured as episodic returns attained when evaluated in the environment. We explore this potentially interesting relationship on a larger scale by adopting a proof by contradiction approach. We evaluated the agents with the hyperparameter setup defined for the Atari datasets in RL Unplugged (Gulcehre et al., 2020) and the hyperparameter sweep

defined for DeepMind Lab (Gulcehre et al., 2021). For a narrow set of hyperparameters this correlation exists as observed in Figure 1. However, in both cases, we notice that a broad hyperparameter sweep makes the correlation between performance and rank disappear (see Figures 1 and DeepMind lab Figure in Appendix A.3). In particular, we find hyperparameter settings that lead to low (collapsed) ranks with high performance (on par with the best performance reported in the restricted hyperparameter range) and settings that lead to high ranks but poor performance. This shows that the correlation between effective rank and performance can not be trusted for offline policy selection. In the following sections, we further present specific ablations that help us understand the dependence of the effective rank vs. performance correlation on specific hyperparameter interventions.

### 4.1 Lifespan of learning with deep Q-networks

Empirically, we found effective rank sufficient to identify three phases when training an offline DQN agent with a ReLU activation function (Figure 2). Although the effective rank may be sufficient to identify those stages, it still does not imply a direct causal link between the effective rank and the performance as discussed in the following sections. Several other factors can confound effective rank, making it less reliable as a guiding metric for offline RL unless those confounders are carefully controlled.

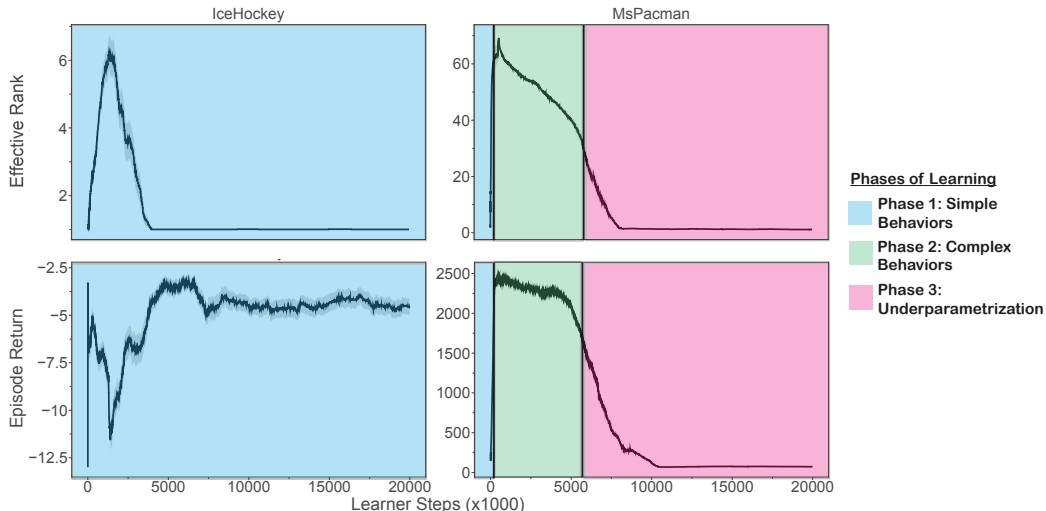

Figure 4: **The three phases of learning:** On IceHockey and MsPacman RL Unplugged Atari games we illustrate the different phases of learning with the offline DQN agent using the learning rate of 0.0004. The blue region in the plots identifies the Phase 1, green region is the Phase 2, and red region is the Phase 3 of the learning. IceHockey is one of the games where the expert that generated the dataset performs quite poorly on it, thus the majority of the data is just random exploration data. It is easy to see that in this phase, the Offline DQN performs very poorly and never manages to get out of the Phase 1. The performance of the agent on IceHockey is poor and the effective rank of the network is low throughout the training. On MsPacman, we can observe all the three phases. The model transition into Phase 2 from Phase 1 quickly, and then followed by the under-fitting regime where the effective rank collapses the agent performs poorly.

We identified three phases of learning according to which stage they appear during training, the performance (returns obtained by the algorithm) when evaluated in the environment, and the effective rank:

- **Phase 1 (Simple behaviors):** These behaviors emerge early in training and give rise to low rewards when evaluated in the environment (often close to a random policy.) The effective rank of the model first collapses to a small value, sometimes to a single-digit value, and then gradually increases. Let us note the term *s*imple behaviors, here, refers to how easy/simple to learn a behavior for an RL algorithm. We hypothesized that this could be due to the implicit bias of the SGD to learn functions of increasing

complexity over training iterations (Kalimeris et al., 2019); therefore, early in training, the network relies on *simple behaviours* that are myopic to most of the variation in the data. Hence the model would have a low rank. The rank collapse early at the beginning of the training happens very abruptly; just in a handful of gradient updates, the effective rank collapses to a single-digit number. However, this early rank collapse does not degrade the agent's performance. We call this rank collapse early in training as *self-pruning effect.*

- **Phase 2 (Complex behaviors):** These behaviors emerge later in training and achieve high rewards, often close to the best policy in the dataset when evaluated in the environment. In this phase, the effective rank of the model first increases and then usually flattens. The model starts to learn more complex behaviors that would achieve high returns when evaluated in the environment. We call these behaviors as *c*omplex behaviors, because of the complexity/difficulty to learn these behaviors since they emerge in later during training and sensitive to the hyperparameters.

- **Phase 3 (Underfitting/Underparameterization):** This is the last phase of the algorithm; in this phase, the effective rank collapses to a small value (often to 1), and the agent's performance collapses too. The third phase is called *underfitting* since the agent's performance usually drops, and the effective rank also collapses, which causes the agent to lose part of its capacity. This phase is not always observed (or the performance does not collapse with effective ran towards the end of the training) in all settings as we demonstrate in our different ablations. Typically in supervised learning, Phase 2 is followed by over-fitting, but with offline TD-learning, we could not find any evidence of over-fitting. We believe this phase is primarily due to the target Q-network needing to extrapolate over the actions not seen during the training and causing extrapolation errors as described by (Kumar et al., 2019). A piece of evidence to support this hypothesis is presented in Figure 29 in Appendix A.5, which suggests that the effective rank and the value error of the agent correlate well. In this phase, the low effective rank and poor performance are caused by a large number of dead ReLU units. Shin and Karniadakis (2020) also show that as the network has an increasing number of dead units, it becomes under-parameterized, and this could influence the agent's performance negatively.

It is possible to identify those three phases in many of the learning curves we provide in this paper, and our first two phases agree with the works on SGD's implicit bias of learning functions of increasing complexity (Kalimeris et al., 2019). Given a fixed model and architecture, whether it is possible to observe all these three phases during training fundamentally depends on:

1. **Hyperparameters:** The phases that an offline RL algorithm would go through during training depends on hyperparameters such as learning rate and the early stopping or training budget. For example, due to early stopping the model may just stop in the second phase, if the learning rate is too small, since the parameters will move much slower the model may never get out of Phase 1. If the model is not large enough, it may never learn to transition from Phase 1 into Phase 2.

2. **Data distribution:** The data distribution has a very big influence on the phases the agent goes thorough, for example, if the dataset only contains random exploratory data and ORL method may fail to learn complex behaviors from that data and as a result, will never transition into Phase 2 from Phase 1.

3. **Learning Paradigm:** The learning algorithm, including optimizers and the loss function, can influence the phases the agent would go through during the training. For example, we observed that Phase 3 only happens with the offline TD-learning approaches.

It is possible to avoid phase three (underfitting) by finding the correct hyperparameters. We believe the third phase we observe might be due to non-stationarity in RL losses (Igl et al., 2020) due to bootstrapping or errors propagating through bootstrapped targets (Kumar et al., 2021b). The underfitting regime only appears if the network is trained long enough. The quality and the complexity of the data that an agent learns from also plays a key role in deciding which learning phases are observed during training. In Figure 4, we demonstrate the different phases of learning on IceHockey and MsPacman games. On IceHockey, since the expert that generated the dataset has a poor performance on that game, the offline DQN is stuck in

Phase 1, and did not managed to learn complex behaviors that would push it to Phase 2 but on MsPacMan, all the three phases are present. We provide learning curves for all the online policy selection Atari games across twelve learning rates in Appendix A.8.

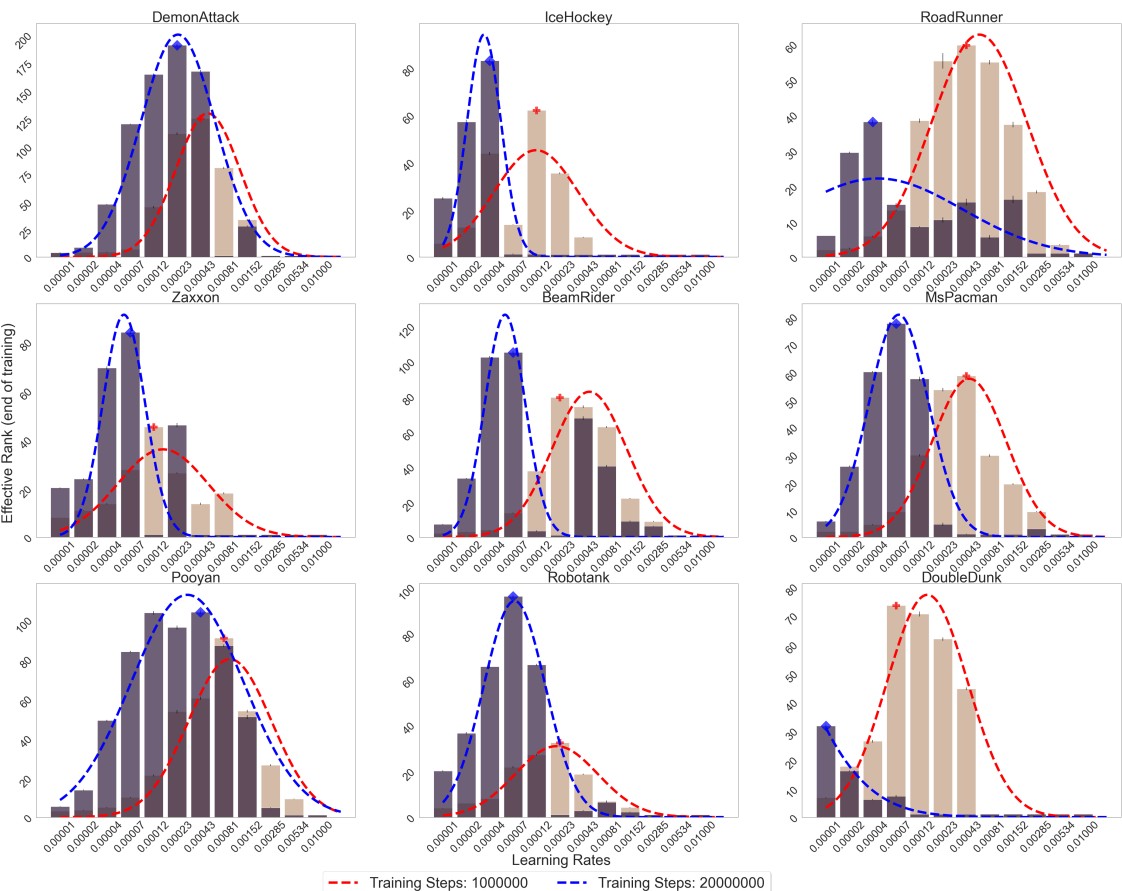

Figure 5: [**Atari**]: Bar charts of effective ranks with respect to the learning rates after 1M and 20M learning steps. After 1M gradient steps, the ranks are distributed almost like a bell-shaped curve, indicating that high and low learning rates have low ranks (phases 1 and 3) while the learning rates in the middle are in phase 2 and hence have the higher rank. After 20M learning steps, the mode of the distribution of the ranks skews towards the left, indicating low terminal ranks for many large learning rates. Namely, as we train the network longer, the terminal rank decreases, particularly for the large learning rates. The rank is low for the low learning rates because the model is stuck in Phase 1, whereas the large learning rates get into Phase 3 quickly and thus the low terminal ranks.

Figure 5 shows the relationship between the effective rank and twelve learning rates. In this figure, the effect of learning rate on the different phases of learning is distinguishable. For low learning rates, the ranks are low because the agent can never transition from Phase 1 to Phase 2, and for large learning rates, the effective ranks are low because the agent is in Phase 3. Therefore, the distribution of effective ranks and learning rates has a Gaussian-like shape, as depicted in the figure. The distribution of rank shifts towards low learning rates as we train the agents longer because slower models with low learning rates start entering Phase 2, and the models trained with large learning rates enter Phase 3.

## 4.2 The effect of dataset size

We can use the size of the dataset as a possible proxy metric for the coverage that the agent observes in the offline data. We uniformly sampled different proportions (from 5% of the transitions to the entire dataset)

from the transitions in the RL Unplugged Atari benchmark dataset (Gulcehre et al., 2020) to understand how the agent behaves with different amounts of training data and whether this is a factor affecting the rank of the network.

Figure 6 shows the evolution of rank and returns over the course of training. The effective rank and performance collapse severely with low data proportions, such as when learning only on 5% of the entire dataset subsampled. Those networks can never transition from phase 1 to phase 2. However, as the proportion of the dataset subsampled increases, the agents could learn more complex behaviors to get into phase 2. The effective rank collapses less severely for the larger proportions of the dataset, and the agents tend to perform considerably better. In particular, we can see that in phase 1, an initial decrease of the rank correlates with an increase in performance, which we can speculate is due to the network reducing its reliance on spurious parts of the observations, leading to representations that generalize better across states. It is worth noting that the ordering of the policies obtained by using the agents' performance does not correspond to the ordering of the policies with respect to the effective rank throughout training. For example, offline DQN trained on the full dataset performs better than 50% of the dataset, while the agent trained using 50% of the data sometimes has a higher rank. A similar observation can be made for Zaxxon, at the end of the training, the network trained on the full dataset underperforms compared to the one trained on 50% of the data, even if the rank is the same or higher.

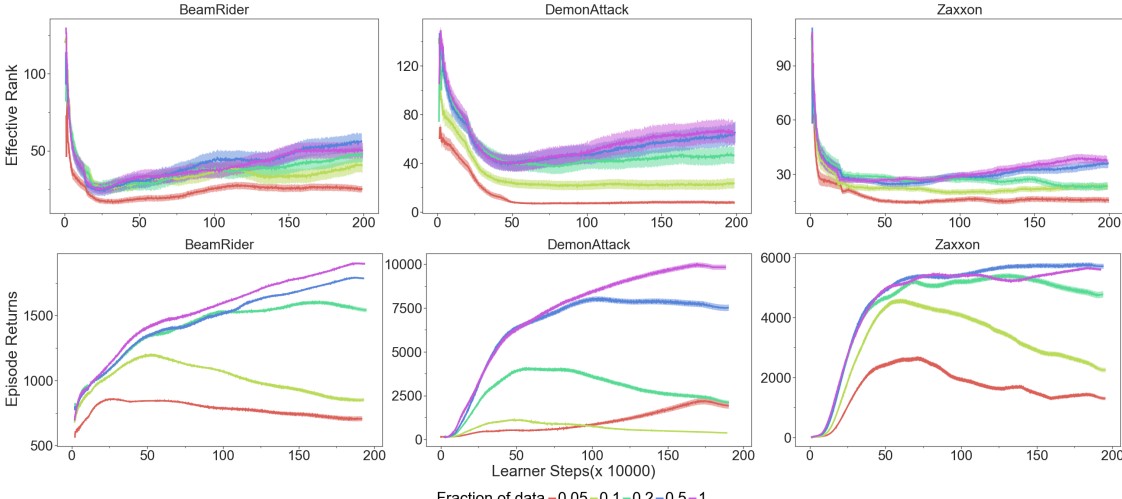

Figure 6: [**Atari**] **Dataset size:** Evolution of ranks and returns as we vary the fraction of data available for the agent to train on. We see that the agent which sees very little data collapses both in terms of rank and performance. The agent which sees more of the data has good performance even while allowing some shrinkage of rank during the training.

## 5  Interactions between rank and performance

To understand the interactions and effects of different factors on the rank and performance of offline DQN, we did several experiments and tested the effect of different hyperparameters on effective rank and the agent's performance. Figure 3 shows the causal graph of the effective rank, its confounders, and the agent's performance. Ideally, we would like to intervene in each node on this graph to measure its effect. As the rank is a continuous random variable, it is not possible to directly intervene on effective rank. Instead, we emulate interventions on the effective rank by conditioning to threshold $\beta$ with $\tau$. In the control case ($\lambda(1)$), we assume $\beta > \tau$ and for $\lambda(0)$, we would have $\beta \leq \tau$.

We can write the average treatment effect (ATE) (Holland, 1986) for the effect of setting the effective rank to a large value on the performance as:

$$\text{ATE}(\lambda, \beta, \tau) = \mathbb{E}[\lambda|\lambda(1)] - \mathbb{E}[\lambda|\lambda(0)]. \tag{3}$$

Let us note that this $\text{ATE}(\lambda, \beta, \tau)$ quantity doesn't necessarily measure the causal effect of $\beta$ on $\lambda$, since we know that the $\lambda$ can be confounded by $\mathbf{A}$ (hidden confounders.)

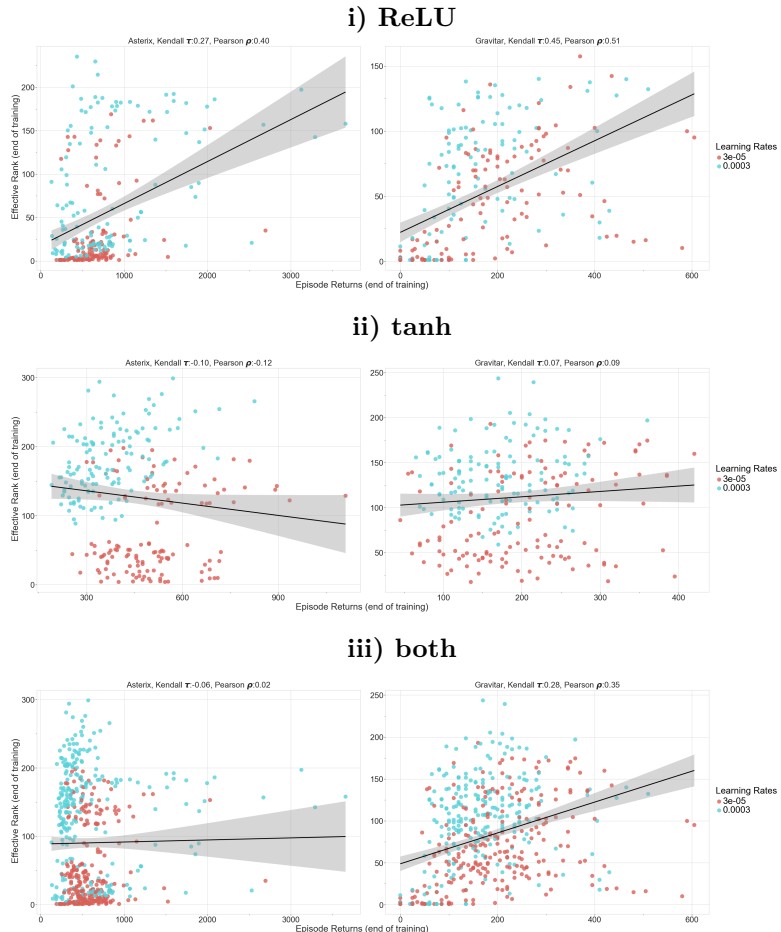

Figure 7: [**Atari**]: The correlation plot between the effective rank and the performance (measured in terms of episode returns by evaluating the agent in the environment) of offline DQN on Asterix and Gravitar games over 256 different hyper-parameter configurations trained for 2M learning steps. There is a strong correlation with the ReLU function, but the correlation disappears for the network with *tanh* activation function. There is no significant correlation between effective rank and the performance on the Asterix game with the complete data. Still, a positive correlation exists on the Gravitar game. These results are not affected by the Simpson's paradox since the subgroups of the data when split into groups concerning activation functions, do not show a consistent correlation trend.

We study the impact of factors such as activation function, learning rate, data split, target update steps, and CURL and SAM losses on the Asterix and Gravitar levels. We chose these two games since Asterix is an easy-to-explore Atari game with relatively dense rewards, while Gravitar is a hard-to-explore and a sparse reward setting (Bellemare et al., 2016). In Table 1, we present the results of intervening to $\beta$ thresholded with different quantiles of the effective rank. Choosing a network with a high effective rank for a ReLU network has a statistically significant positive effect on the agent's performance concerning different quantiles on both Asterix and Gravitar. The agent's performance is measured in terms of normalized scores as described in Gulcehre et al. (2020). However, the results with the tanh activation function are mixed, and the effect of changing the rank does not have a statistically significant impact on the performance in most cases. In Figure 7, we show the correlations between the rank and performance of the agent. The network with ReLU activation strongly correlates rank and performance, whereas *tanh* does not. *The experimental data can be prone to Simpson's paradox which implies that a trend (correlation in this case) may appear in several*

*subgroups of the data, but it disappears or reverses when the groups are aggregated (Simpson, 1951). This can lead to misleading conclusions.* We divided our data into equal-sized subgroups for hyperparameters and model variants, but we could not find a consistent trend that disappeared when combined, as in Figure 7. Thus, our experiments do not appear to be affected by Simpson's paradox.

Table 1: **Atari:** The average treatment effect of having a network with a higher rank than concerning different quantiles. We report the average treatment effect (ATE), its uncertainty (using 95% confidence intervals using standard errors), and p-values for Asterix and Gravitar games. Higher effective rank seems to have a smaller effect on Asterix than Gravitar. Let us note that, Gravitar is a sparse reward problem, and Asterix is a dense-reward one. Overall the effect is more prominent for ReLU than tanh, and with tanh networks, it is not statistically significant. We boldfaced the ATEs where the effect is statistically significant ($p < 0.05$.) The type column of the table indicates the activation functions used in the experiments we did the intervention. The "Combined" type corresponds to a combination of tanh and ReLU experiments.

| Level | Type | quantile=0.25 | | quantile=0.5 | | quantile=0.75 | | quantile=0.95 | |
|---|---|---|---|---|---|---|---|---|---|
| | | ATE | p-value | ATE | p-value | ATE | p-value | ATE | p-value |
| Asterix | Combined | **0.019 ± 0.010** | **0.001** | 0.007 ± 0.013 | 0.207 | 0.005 ± 0.019 | 0.324 | -0.031 ± 0.011 | 0.999 |
| | ReLU | **0.079 ± 0.018** | **0.000** | **0.089 ± 0.025** | **0.000** | **0.112 ± 0.040** | **0.000** | **0.110 ± 0.085** | **0.017** |
| | Tanh | -0.014 ± 0.004 | 0.999 | 0.009 ± 0.005 | 0.996 | -0.005 ± 0.006 | 0.886 | **0.020 ± 0.017** | **0.023** |
| Gravitar | Combined | **0.954 ± 0.077** | **0.000** | **0.569 ± 0.098** | **0.000** | **0.496 ± 0.141** | **0.000** | **0.412 ± 0.206** | **0.001** |
| | ReLU | **1.654 ± 0.150** | **0.000** | **1.146 ± 0.163** | **0.000** | **1.105 ± 0.256** | **0.000** | **1.688 ± 0.564** | **0.000** |
| | Tanh | 0.028 ± 0.090 | 0.303 | **0.185 ± 0.118** | **0.005** | **0.242 ± 0.167** | **0.009** | 0.054 ± 0.265 | 0.367 |

Table 2: [**Atari**] **Average Treatment Effect** of different interventions on Asterix and Gravitar: Quantity of interest $Y$ is the terminal rank of the features. Activation function and learning rate influence the terminal feature ranks the most in our setup. We indicate them with boldface; changing the activation from ReLU to tanh improves the effective rank, whereas reducing the learning rate reduces the rank.

| u | Control (**u**) | Treatment **T(u)** | $Y_{\mathbf{t(u)}} - Y_{\mathbf{c(u)}}$ |
|---|---|---|---|
| Activation Function | ReLU | tanh | **66.97 ± 7.13** |
| Learning Rate | $3 \times 10^{-4}$ | $3 \times 10^{-5}$ | **-54.15 ± 7.54** |
| Target Update Steps | 2500 | 200 | -15.23 ± 8.21 |
| SAM Loss Weight | 0 | 0.05 | -10.05 ± 8.26 |
| Level | Asterix | Gravitar | -9.41 ± 8.25 |
| CURL Loss Weight | 0 | 0.001 | 8.09 ± 8.27 |
| Data Split | 100% | 5% | 5.11 ± 8.27 |

In this analysis, we set our control setting to be trained with ReLU activations, the learning rate of $3e-4$, without any auxiliary losses, with training done on the full dataset in the level Asterix. We present the *Average Treatment Effect* (ATE) (the difference in ranks between the intervention being present and absent in an experiment) of changing each of these variables in Table 2. We find that changing the activation function from ReLU to tanh drastically affects the effective ranks of the features, which is in sync with our earlier observations on other levels. In Figure 8, we present a heatmap of ATEs where we demonstrate the effective rank change when two variables are changed from original control simultaneously. Once again, the activation functions and learning rate significantly affect the terminal ranks. We also observe some interesting combinations that lead the model to converge to a lower rank – for example, using SAM loss with dropout.

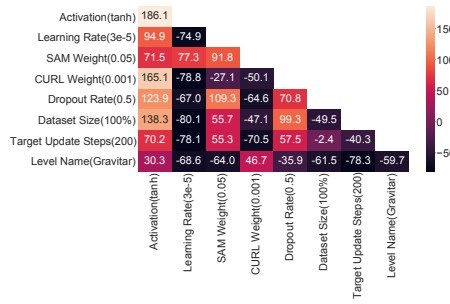

Figure 8: [**Atari**] **ablations:** Effects of different pairs of interventions on the control set. The control is on Asterix without dropout, CURL and SAM on the full dataset and with a target update period of 2500 steps.

These observations further reinforce our belief that different factors affect the phenomenon. The extent of rank collapse and mitigating rank collapse alone may never fully fix the agent's learning ability in different environments.

## 6  The effect of activation functions

The severe rank collapse of phase 3 is apparent in our Atari models, which have simple convolutional neural networks with ReLU activation functions. When we study the same phenomenon on the SeekAvoid dataset, rank collapse does not seem to happen similarly. It is important to note here that to solve those tasks effectively, the agent needs memory; hence, all networks have a recurrent core and LSTM. Since standard LSTMs use $tanh(\cdot)$ activations, investigating in this setting would help us understand the role of the choice of architecture on the behavior of the model's rank.

Figure 10 shows that the output features of the LSTM network on the DeepMind lab dataset do not experience any detrimental effective rank collapse with different exploration noise when we use $tanh(\cdot)$ activation function for the cell. However, if we replace the $tanh(\cdot)$ activation function of the LSTM cell with ReLU or if we replace the LSTM with a feed-forward MLP using ReLU activations (as seen in Figure 10) the effective rank in both cases, collapses to a small value at the end of training. This behavior shows that the choice of activation function has a considerable effect on whether the model's rank collapses throughout training and, subsequently, its ability to learn expressive value functions and policies in the environment as it is susceptible to enter phase 3 of training.

> **Observation:** Agents that have networks with ReLU units tend to have dead units which causes the effective rank to collapse in phase 3 of learning while other activations like tanh do not suffer a similar collapse.

The activation functions influence both the network's learning dynamics and performance. As noted by Pennington and Worah (2017), the activation function can influence the rank of each layer at initialization. Figure 9 presents our findings on bsuite levels. In general, the effective rank of the penultimate layer with ReLU activations collapses faster, while ELU and $tanh(\cdot)$ tend to maintain a relatively higher rank than ReLU. As the effective rank goes down for the catch environment, the activations become sparser, and the units die. We illustrate the sparsity of the activations with Gram matrices over the features of the last hidden layer of a feedforward network trained on bsuite in Figure 37 in Appendix A.13.

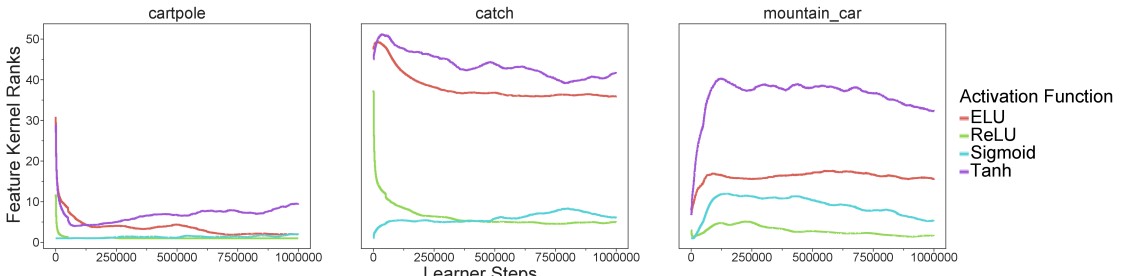

Figure 9: [**bsuite**] **Effective ranks of different activation functions:** The magnitude of drop in effective rank is more severe for **ReLU** and **sigmoid** activation functions than **tanh**.

## 7  Optimization

The influence of minibatch size and the learning rate on the learning dynamics is a well-studied phenomenon. Smith and Le (2017); Smith et al. (2021) argued that the ratio between the minibatch size and learning rate relates to implicit regularization of SGD and also affects the learning dynamics. Thus, it is evident that

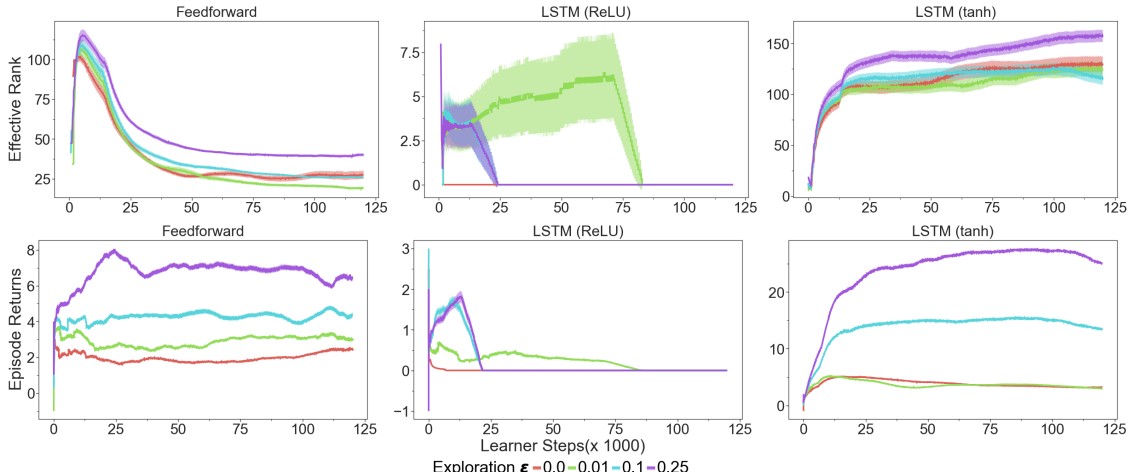

Figure 10: [**DeepMind lab SeekAvoid**] **Activation functions:** Evolution of ranks in a typical LSTM network with **tanh** activations at the cell of LSTM on DeepMind lab. We see that a typical LSTM network does not get into Phase 3. However, when we change the activation function of the cell gate from **tanh** to **ReLU** the effective rank collapses to very small values. The effective rank collapses when the LSTM is replaced with a feedforward network too in cases of ReLU activation.

those factors would impact the effective rank and performance. Here, we focus on a setup where we see the correlation between effective rank and performance and investigate how it emerges.

Our analysis and ablations in Table 2 and Figure 8 illustrate that the learning rate is one of the most prominent factors on the performance of the agent. In general, there is no strong and consistent correlation between the effective rank and the performance across different models and hyperparameter settings. However, in Figure 1, we showed that the correlation exists in a specific setting with a particular minibatch size and learning rate. Further, in Figure 7, we narrowed down that the correlation between the effective rank and the performance exists for the offline DQN with ReLU activation functions. Thus in this section, we focus on a regime where this correlation exists with the offline DQN with ReLU and investigate how the learning rate and minibatches affect the rank. We ran several experiments to explore the relationship between the minibatch size, learning rates, and rank. In this section, we report results on Atari in few different settings – Atari-DQN-256-2M, Atari-DQN-32-100M, Atari-DQN-256-20M and Atari-BC-256-2M.

The dying ReLU problem is a well-studied issue in supervised learning (Glorot et al., 2011; Gulcehre et al., 2016) where due to the high learning rates or unstable learning dynamics, the ReLU units can get stuck in the zero regime of the ReLU activation function. We compute the number of dead ReLU units of the penultimate layer of a network as the number of units with zero activation value for all inputs. Increasing the learning rate increases the number of dead units as can be seen in Figures 11 and 12. Even in BC, we observed that using large learning rates can cause dead ReLU units, rank collapse, and poor performance, as can be seen in Figure 13, and hence this behavior is not unique to offline RL losses. Nevertheless, models with TD-learning losses have catastrophic rank collapses and many dead units with lower learning rates than BC. Let us note that the effective rank depends on the number of units ($D$) and the number of dead units ($\eta$) at a layer. It is easy to see that effective rank is upper-bounded by $D - \eta$. In Figure 14, we observe a strong correlation between the number of dead ReLU units of the penultimate layer of the ReLU network and its effective rank.

> **Observation:** The pace of learning influences the number of dead units and the effective rank: the larger the learning rates and the smaller minibatches are, the number of dead units increases, and the effective rank decreases. However, the performance is only poor when the effective rank is severely low.

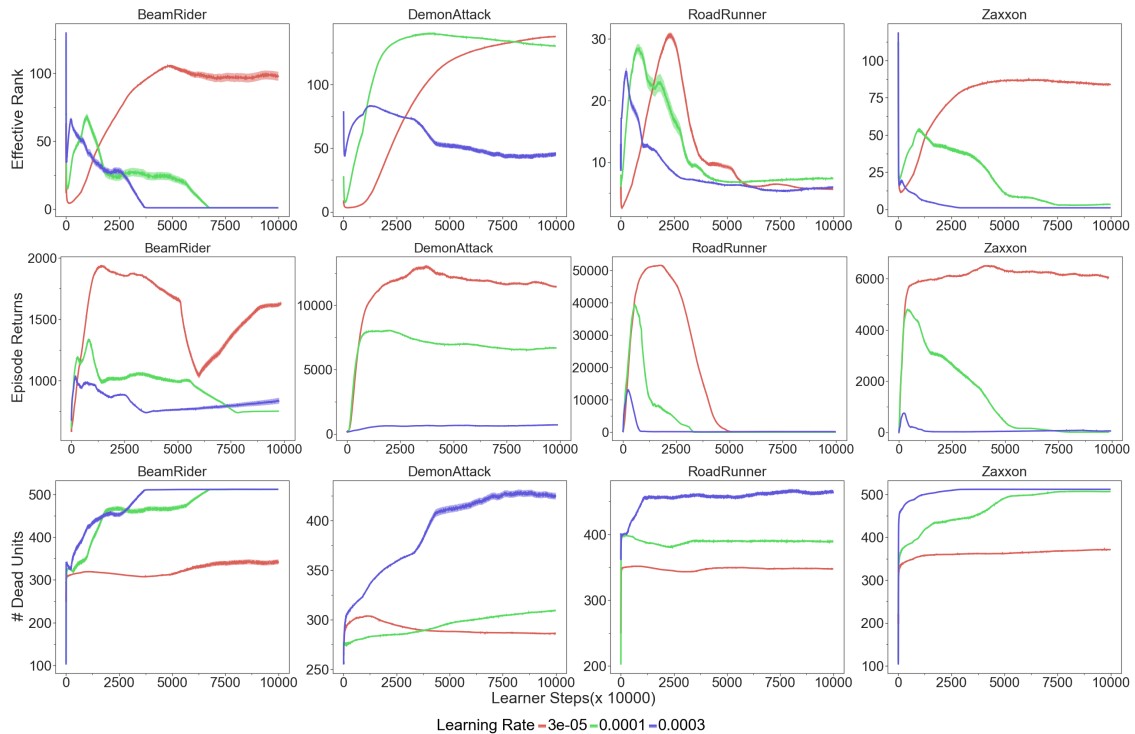

Figure 11: [**Atari DQN-32-100M**] **Learning curves of offline DQN** for 100M gradient steps of training with minibatch size of 32. Increasing the learning rate increases the number of dead units in the ReLU network. As a result, the increased learning rate also causes more severe collapse as well, which aligns very well with the number of dead units. We observed that this behavior can also happen with networks using saturating activation functions such as sigmoid.

Finally, we look into how effective ranks shape up towards the end of training. We test 12 different learning rates, trying to understand the interaction between the learning rates and the effective rank of the representations. We summarize our main results in Figure 5 where training offline DQN decreases the effective rank. The effective rank is low for both small and large learning rates. For higher learning rates, as we have seen earlier, training for longer leads to many dead ReLU units, which in turn causes the effective rank to diminish, as seen in Figures 11 and 12. Moreover, as seen in those figures, in Phase 1, the effective rank and the number of dead units are low. Thus, the rank collapse in Phase 1 is not caused by the number of dead units. In Phase 3, the number of dead units is high, but the effective rank is drastically low. The drastically low effective rank is caused by the network's large number of dead units in Phase 3. In Phase 3, we believe that both the under-parameterization and the poor performance is caused by the number of dead units in ReLU DQN, which was shown to be the case by (Shin and Karniadakis, 2020) in supervised learning.

Overall, we could only observe a high correlation between the effective rank and the agent's performance, when we use ReLU activations in the network after a long training. We also present more analysis on how controlling the loss landscape affects the rank vs performance in Appendix A.2 and A.4.

## 8 The effect of loss function

### 8.1 Q-learning and behavior cloning

Behavior cloning (BC) is a method to learn the behavior policy from an offline dataset using supervised learning approaches (Pomerleau, 1989). Policies learned by BC will learn to mimic the behavior policy, and thus the performance of the learned BC agent is highly limited by the quality of the data the agent is trained on. We compare BC and Q-learning in Figure 15. We confirm that with default hyperparameters, the BC

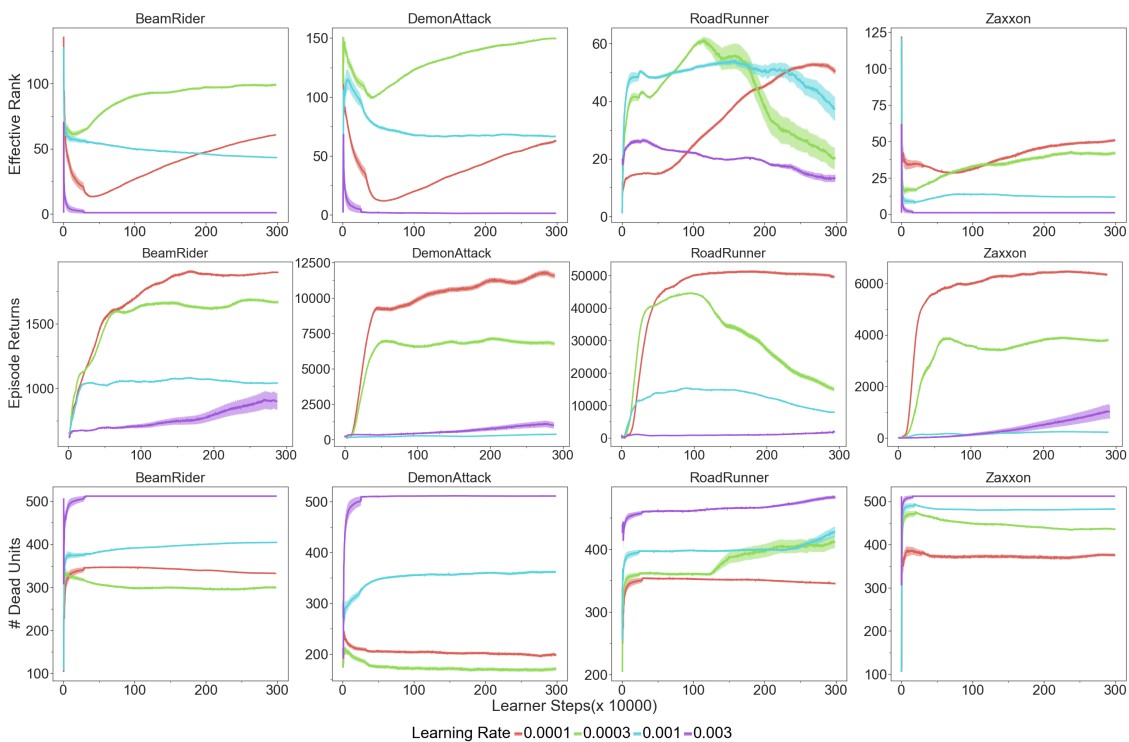

Figure 12: [**Atari DQN-256-3M**] **Learning curves of offline DQN** for 3M gradient steps of training with minibatch size of 256. Increasing the minibatch size improves the performance of the network with larger learning rates.

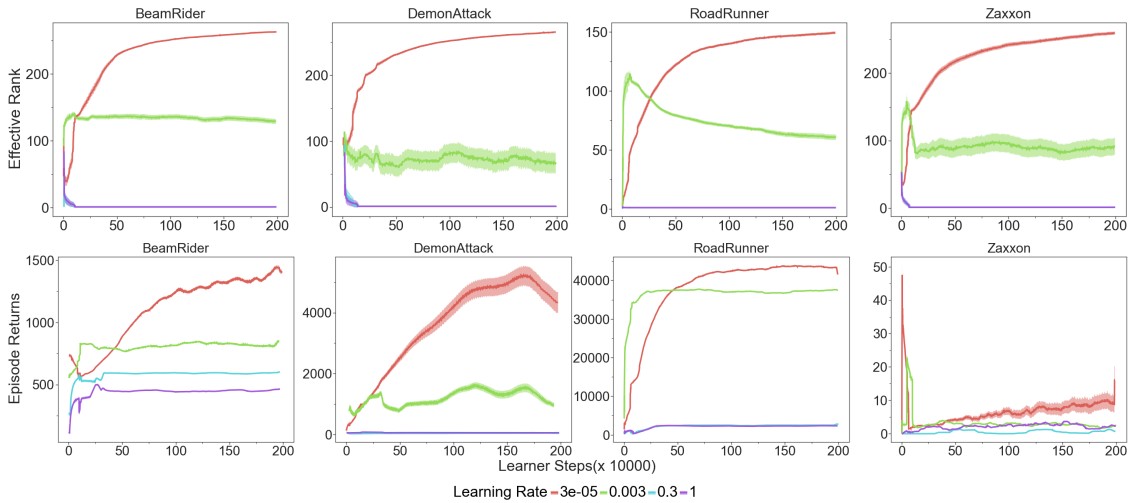

Figure 13: [**Atari BC-256-2M**] **Learning Curves of BC** After 2M gradient steps with mini-batch size of 256. For large enough learning rates the rank of BC agent also collapses. We hypothesize that the rank collapse is a side effect of learning in general and not only due to TD-learning based losses.

agent's effective rank does not collapse at the end of training. In contrast, as shown by Kumar et al. (2020a), DQN's effective rank collapses. DQN outperforms BC even if its rank is considerably lower, and during learning, the rank is not predictive of the performance (Figure 15). This behavior indicates the unreliability of effective rank for offline model selection.

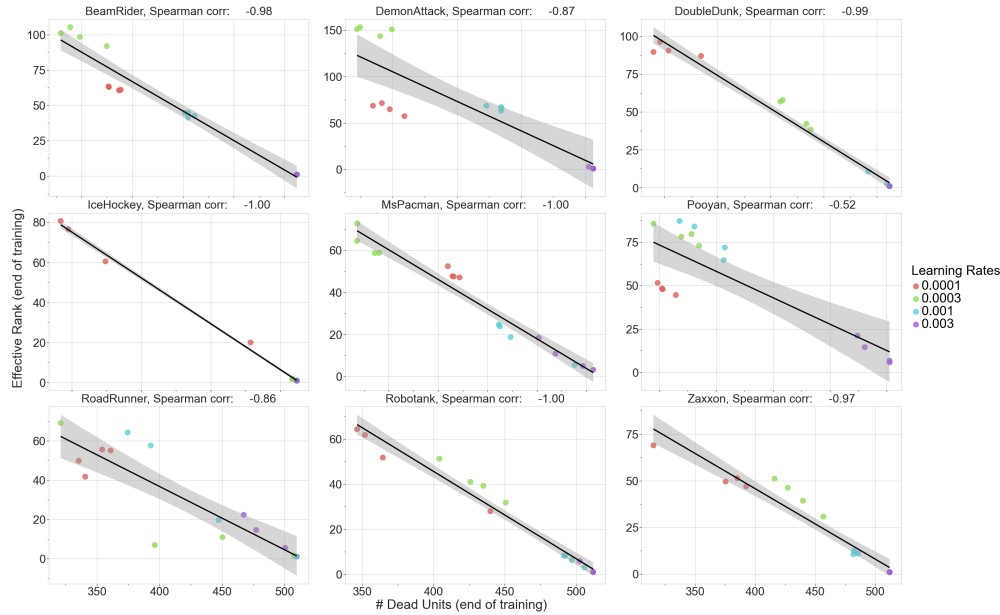

Figure 14: [**Atari**] Scatter plot for the correlation between the number of dead units and effective rank at the end of training and we observe that the effective rank strongly correlates with the number of dead units. Also, using larger learning rate increases the number of dead units in the network.

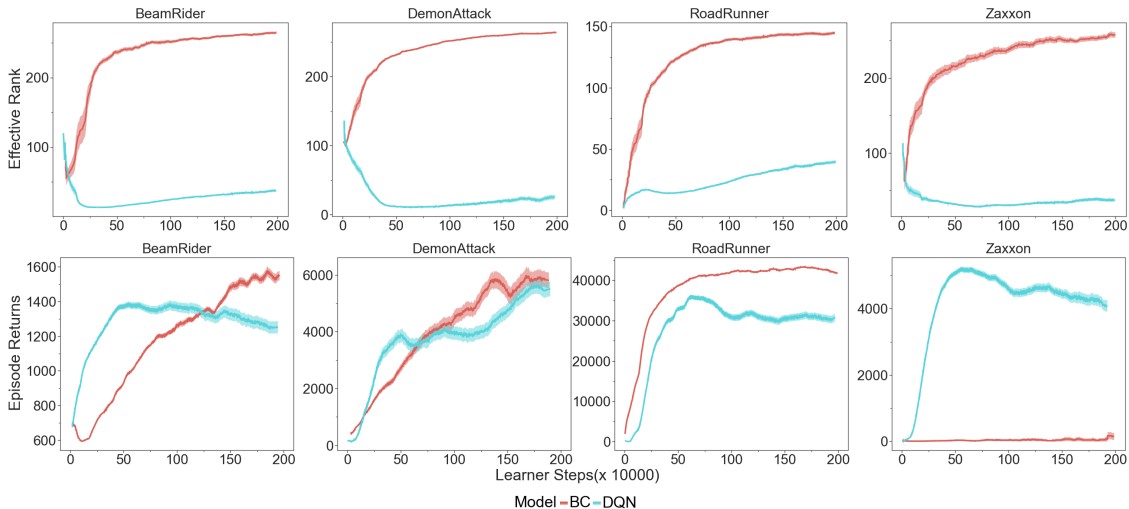

Figure 15: [**Atari**] **Offline DQN and Behavior Cloning (BC):** We compare BC and DQN agents on the Atari dataset. We used the same architecture, dataset, and training protocols for both baselines. We used the hyperparameters defined in (Gulcehre et al., 2020) for comparisons. The rank of the DQN agent is significantly lower and achieves higher returns than BC.

## 8.2   CURL: The effect of self-supervision

We study whether adding an auxiliary loss term proposed for CURL (Laskin et al., 2020) (Equation 2) during training helps the model mitigate rank collapse. In all our Atari experiments, we use the CURL loss described in (Laskin et al., 2020) without any modifications. Since the DeepMind lab tasks require memory, we apply a similar CURL loss to the features aggregated with a mean of the states over all timesteps. In all experiments, we also sweep over the weight of the CURL loss $\lambda$.

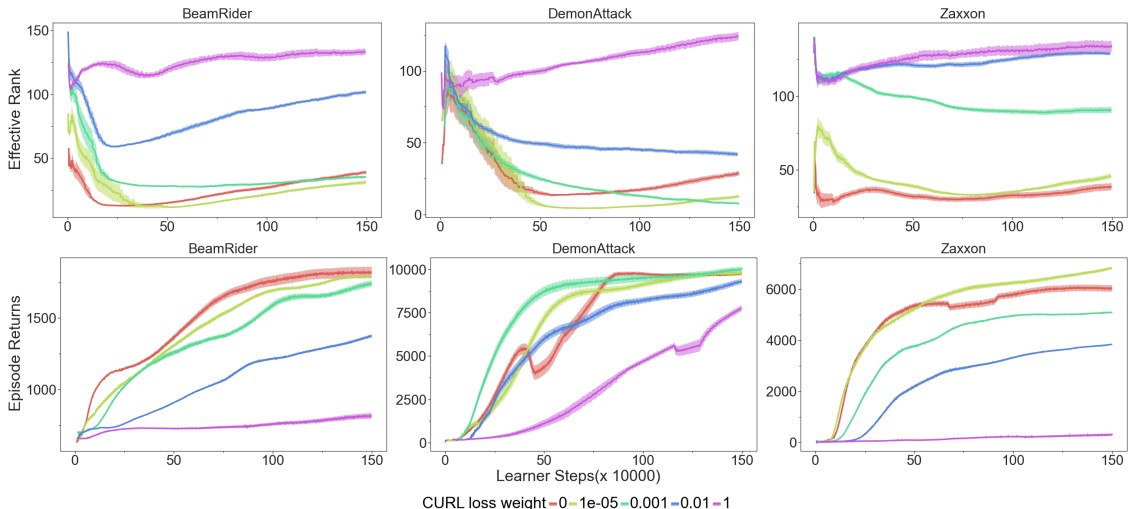

Figure 16: [**Atari**] **Auxiliary losses:** Evolution of ranks as we increase the weight of auxiliary losses. We see that a strong weight for auxiliary loss helps mitigate the rank collapse but prevents the model from learning useful representations.

Figure 16 shows the ranks and returns on Atari (for DeepMind lab results see Appendix A.6.) In Atari games, using very large weights for an auxiliary loss ($\approx 1$) prevents rank collapse, but they simultaneously deteriorate the agent performance. We speculate, borrowing on intuitions from the supervised learning literature on the role of the rank as implicit regularizer Huh et al. (2021), that in such scenarios large rank prevents the network from ignoring spurious parts of the observations, which affects its ability to generalize. On the other hand, moderate weights of CURL auxiliary loss do not significantly change the rank and performance of the agent. Previously Agarwal et al. (2021) showed that the CURL loss does not improve the performance of an RL agent on Atari in a statistically meaningful way.

On DeepMind lab games, we do not observe any rank collapse. In none of our DeepMind lab experiments agents enter into Phase 3 after Phase 1 and 2. This is due to the use of the tanh activation function for LSTM based on our investigation of the role of the activation functions in Section 6.

## 9   Tandem RL

Kumar et al. (2020a) propose one possible hypothesis for the observed rank collapse as the re-use of the same transition samples multiple times, particularly prevalent in the offline RL setting. A setting in which this hypothesis can be tested directly is the 'Tandem RL' proposed in Ostrovski et al. (2021): here a secondary ('passive') agent is trained from the data stream generated by an architecturally equivalent, independently initialized baseline agent, which itself is trained in a regular, online RL fashion. The passive agent tends to under-perform the active agent, despite identical architecture, learning algorithm, and data stream. This setup presents a clean ablation setting in which both agents use data in the same way (in particular, not differing in their re-use of data samples), and so any difference in performance or rank of their representation cannot be directly attributable to the reuse of data.

In Figure 17, we summarize the results of a Tandem-DQN using Adam (Kingma and Ba, 2014) and RMSProp (Tieleman et al., 2012) optimizers. Despite the passive agent reusing data in a similar fashion as the online agent, we observe that it collapses to a lower rank. Besides, the passive agent's performance tends to be significantly (in most cases, catastrophically) worse than the online agent that could not satisfactorily explained just by the extent of difference in their effective ranks alone. We think that the Q-learning algorithm seems to be not efficient enough to exploit the data generated by the active agent to learn complex behaviors that would put the passive agent into Phase 2. We also noticed that, although Adam achieves

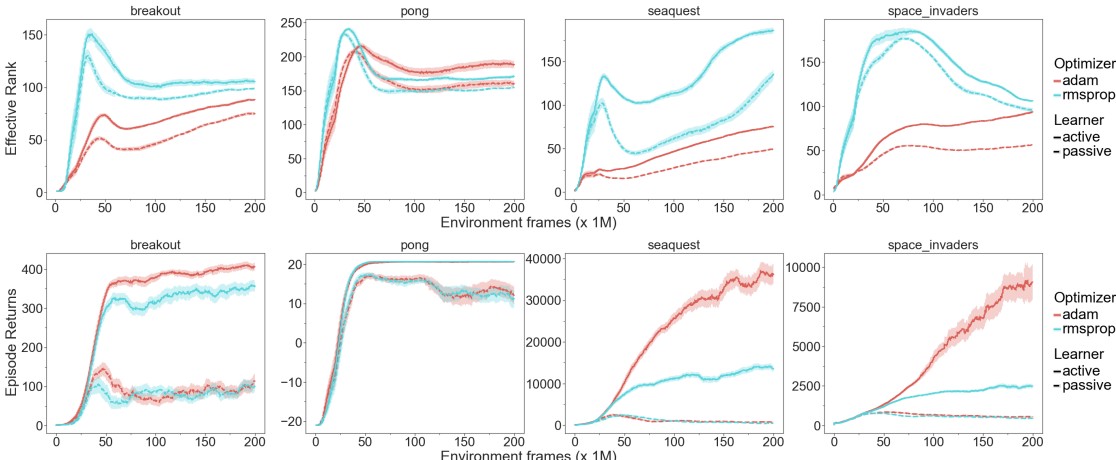

Figure 17: **Atari, tandem RL:** We investigate the effect of the choice of optimizers between Adam and RMSProp on rank and the performance of the models, both for the active (solid lines) and passive agents (dashed lines). We observe the rank of the passive agent is lower than the active agent both for Adam and RMSProp.

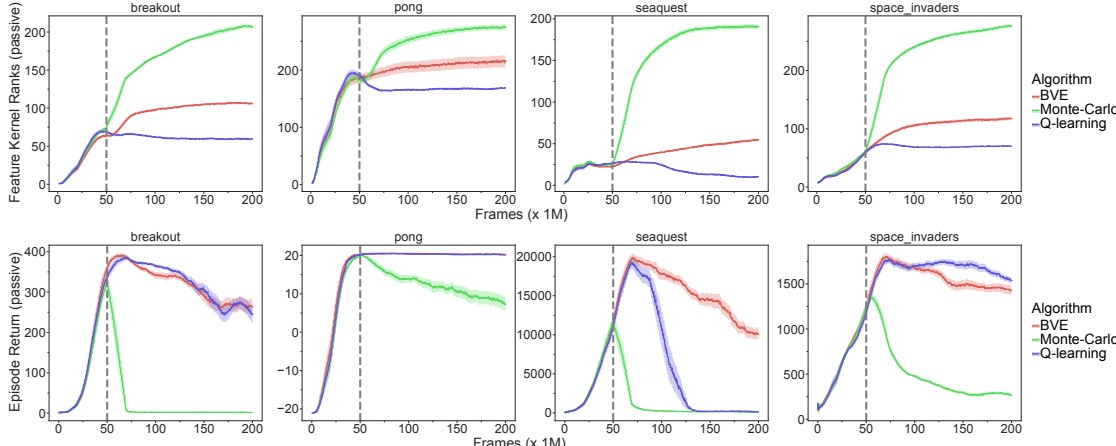

Figure 18: **Atari, Forked tandem RL:** We evaluate different loss functions in the forked tandem setting, where the passive agent is forked from the online agent, and the online agent's parameters are frozen. Still, the parameters of the online agent are updated. We forked the passive agent after it had seen 50M frames during training which is denoted with dashed lines in the figure. We observe that using both BVE and MC return losses improves the agent's rank, but the agent's performance is still poor.

better performance than RMSProp, the rank of the model trained with the Adam optimizer tends to be lower than the models trained with RMSProp.

In Figure 18, we investigate the effect of different learning algorithms on the rank and the performance of an agent in the *forked tandem RL* setting. In the forked tandem setting, an agent is firstly trained for a fraction of its total training time. Then, the agent is 'forked' into active and passive agents, and they both start with the same network weights where the active agent is 'frozen' and not trained but continues to generate data from its policy to train the passive agent on this generated data for the remaining of the training time. Here, we see that the rank flat-lines when we fork the Q-learning agent, but the performance collapses dramatically. In contrast, despite the rank of BVE and an agent trained with on-policy Monte Carlo returns going up, the performance still drops. Nevertheless, the decline of performance for BVE and the agent trained with Monte Carlo returns is not as bad as the Q-learning agent on most Atari games.

## 10 Offline policy selection

In Section 7, we found that with the large learning rate sweep setting rank and number of dead units have high correlation with the performance. Offline policy selection (Paine et al., 2020) aims to choose the best policy without any online interactions purely from offline data. The apparent correlation between the rank and performance raises the natural question of how well rank performs as an offline policy selection approach. We did this analysis on DQN-256-20M experiments where we previously observed strong correlation between the rank and performance. We ran the DQN network described in DQN-256-20M experimental setting until the end of the training and performed offline policy selection using the effective rank to select the best learning rate based on the learning rate that yields to maximum effective rank or minimum number of dead units.

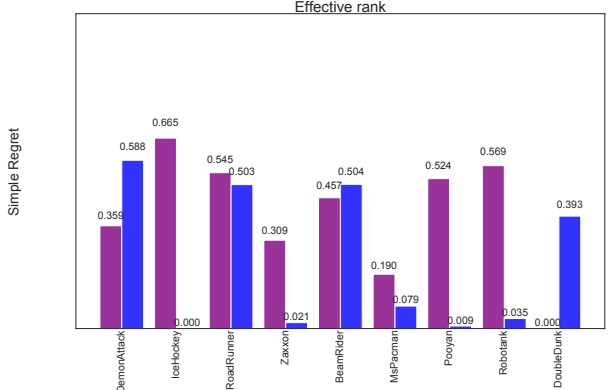
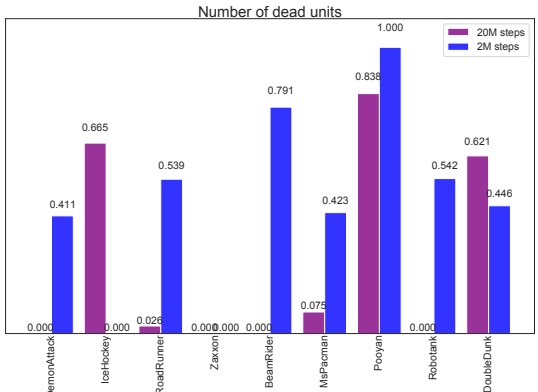

Figure 19: **Atari DQN-256-20M:** This plot is depicting the simple regret of a DQN agent with ReLU activations using effective rank and the number of dead units. On the left, we show the the simple regret to select the best learning rate using the effective rank and on the right hand side, we show the simple regret achieved based on the number of dead units.

Figure 19 illustrates the simple regret on each Atari offline policy selection game. The simple regret between the recommended policy measures how close an agent is to the best policy, and it is computed as described in Konyushkova et al. (2021). A simple regret of 1 would mean that our offline policy selection mechanism successfully chooses the best policy, and 0 would mean that it would choose the worst policy. After 2M training steps, the simple regret with the number of dead units as a policy selection method is poor. In contrast, the simple regret achieved by selecting the agent with the highest effective rank is good. The mean simple regret achieved by using the number of dead units as an offline policy selection (OPS) method is $0.45 \pm 0.11$, where the uncertainty $\pm 0.11$ is computed as standard error across five seeds. In contrast, simple regret achieved by using effective rank as an OPS method is $0.24 \pm 0.12$. The effective ranks for most learning rates collapse as we train longer since more models enter Phase 3 of learningthe number of dead units increases. After 20M of learning steps, the mean simple regret computed using effective rank as the OPS method

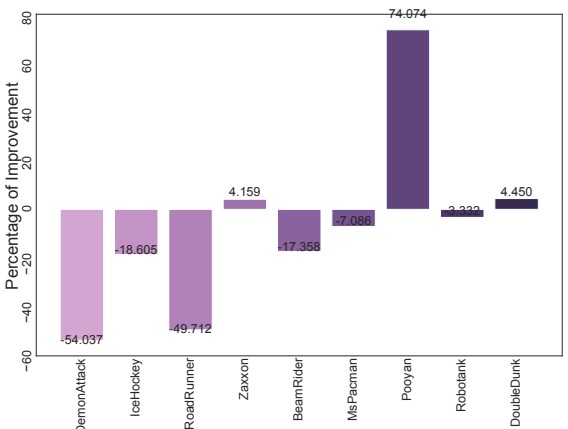

Figure 20: **Atari DQN-256-20M:** Here, we are depicting the percentage of improvement by doing offline policy selection using rank individually vs online policy selection using median reward across nine Atari games to select the best learning rate. Using effective rank as the offline policy selection method performs relatively well when compared to doing online policy selection based on median normalized score across Atari games.

becomes $0.40 \pm 0.07$, and the mean simple regret is $0.25 \pm 0.12$ with the number of dead units for the OPS. Let us note that the 2M learning step is more typical in training agents on the RL Unplugged Atari dataset. The number of dead units becomes a good metric to do OPS when the network is trained for long (20M steps in our experiment), where the rank becomes drastically small for most learning rate configurations. However, effective rank seems to be a good metric for the OPS earlier in training. Nevertheless, without prior information about the task and agent, it is challenging to conclude whether the number of dead units or effective rank would be an appropriate OPS metric. Other factors such as the number of training steps, activation functions, and other hyperparameters confound the effective rank and number of dead units. Thus, we believe those two metrics we tested are unreliable in general OPS settings without controlling those other extraneous factors.

In Figure 20, we compare effective rank as an OPS method with respect to its percentage improvement over the online policy selection to maximize the median reward achieved by the each agent across nine Atari games. The effective rank on Zaxxon, BeamRider, MsPacman, Pooyan, Robotank, DoubleDunk perform competitively to the online policy selection. Effective rank may be a complementary tool for networks with ReLU activation function and we believe it can be a useful metric to monitor during the training in addition to number of dead units to have a better picture about the performance of the agent.

## 11    Robustness to input perturbations

Several works, such as Sanyal et al. (2018) suggested that low-rank models can be more robust to input perturbations (specifically adversarial ones). It is difficult to just measure the effect of low-rank representations on the agent's performance since rank itself is not a robust measure, as it depends on different factors such as activation function and learning which in turn can effect the generalization of the algorithm independently from rank. However, it is easier to validate the antithesis, namely *"Do more robust agents need to have higher effective rank than less robust agents?"*. We can easily test this hypothesis by comparing DQN and BC agents.

**Robustness metric:**   We define the robustness metric $\rho(p)$ based on three variables: i) the noise level $p$, ii) the noise distribution $d(p)$, and iii) the score obtained by agent when evaluated in the environment with noise level $p$: $score[d(p)]$ for which $score[d(0)]$ represents the average score achieved by the agent without any perturbation applied on it. Then we can define $\rho(p)$ as follows:

$$\rho(p) = 1 - \frac{score[d(0)] - score[d(p)]}{score[d(0)]} = \frac{score[d(p)]}{score[d(0)]}. \tag{4}$$

In our experiments, we compare DQN and BC agents since we already know that BC has much larger ranks across all Atari games than DQN. We trained these agents on datasets without any data augmentation. The data augmentations are only applied on the inputs when the agent is evaluated in the environment. We evaluate our agents on BeamRider, IceHockey, MsPacman, DemonAttack, Robotank and RoadRunner games from RL Unplugged Atari online policy selection games. We excluded DoubleDunk, Zaxxon and Pooyan games since BC agent's performance on those levels was poor and very close to the performance of random agent, so the robustness metrics on those games would not be very meaningful. On IceHockey, the results can be negative, thus we shifted the scores by 20 to make sure that they are non-negative. In our robustness experiments, we used the hyperparameters that were used in Gulcehre et al. (2020) on Atari for both for DQN and BC.

**Robustness to random shifts:**   In Figure 21, we investigate the Robustness of DQN and BC agents with respect to random translation image perturbations applied to the observations as described by Yarats et al. (2020). We evaluate the agent on the Atari environment with varying degrees of input scaling. We noticed that BC's performance deteriorates abruptly, whereas the performance of DQN, while also decreasing, is better than the BC one.

**Robustness to random scaling:**   In Figure 22, we explore the Robustness of DQN and BC agents with respect to random scaling as image perturbation method applied to the observations that are feed into our

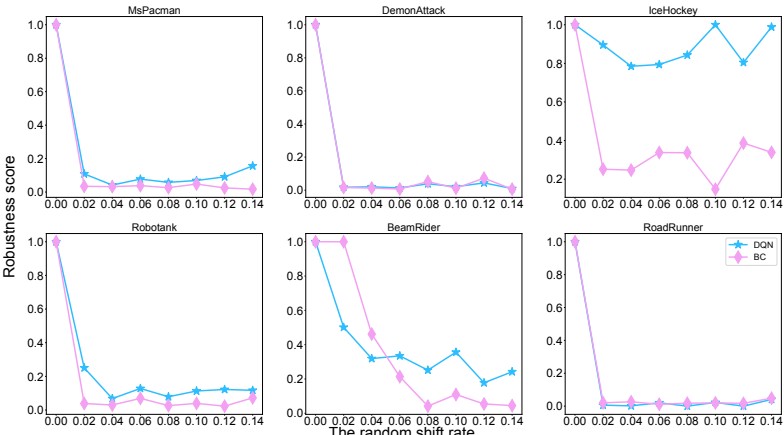

Figure 21: [**Atari**] **Robustness to random shifts during evaluation:** We measure the robustness of BC and DQN to random shifts on six Atari games from RL Unplugged. The DQN and BC agents are trained offline without any perturbations on the inputs, only on the RL Unplugged datasets. However, during evaluation time we perturbed the input images with "random shift" data augmentation. Overall, DQN is more robust than BC to the evaluation-time random-scaling image perturbations that are not seen during training. The difference is more pronounced on IceHockey and BeamRider games. DQN achieved mean AUC (over different games) of 2.042 and BC got 1.26.

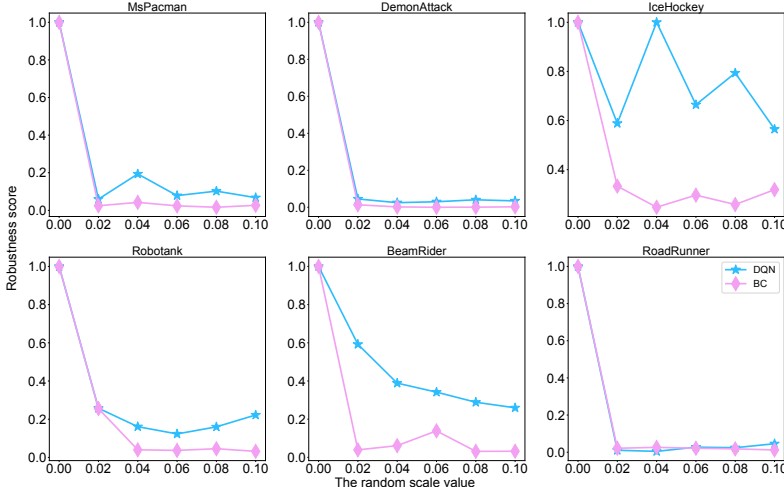

Figure 22: [**Atari**] **Robustness to random scaling during evaluation:** We measure the robustness of BC and DQN to random scaling over inputs on six Atari games from RL Unplugged. The DQN and BC agents are trained offline without data augmentations over the RL Unplugged datasets. However, we perturbed the observations during the evaluation with "random scale" data augmentation. We observed that DQN is more robust than BC to the evaluation-time random-scaling data augmentation. The difference is more pronounced in IceHockey, BeamRider, and Robotank games. DQN achieved a mean AUC (over different games) of 1.60, and BC achieved 0.87.

deep Q-network. We randomly scale the image inputs as described in Chen et al. (2017). When evaluating the agent in an online environment, we test it with varying levels of image scaling. The results are again very similar to the random shift experiments where the DQN agent is more robust to random perturbations in the online environment.

According to the empirical results obtained from those two experiments, it is possible to see that DQN is more robust to the evaluation-time input perturbations, specifically random shifts and scaling than the BC agent. Thus, more robust representations do not necessarily require a higher effective rank.

## 12    Discussion

In this work, we empirically investigated the previously hypothesized connection between low effective rank and poor performance. We found that the relationship between effective rank and performance is not as simple as previously conjectured. We discovered that an offline RL agent trained with Q-learning during training goes through three phases. The effective rank collapses to severely low values in the first phase –we call this as the self-pruning phase– and the agent starts to learn basic behaviors from the dataset. Then in the second phase, the effective rank starts going up, and in the third phase, the effective rank collapses again. Several factors such as learning rate, activation functions and the number of training steps, influence the occurrence, persistence and the extent of severity of the three phases of learning.

In general, a low rank is not always indicative of poor performance. Besides strong empirical evidence, we propose a hypothesis trying to explain the underlying phenomenon: not all features are useful for the task the neural network is trying to solve, and low rank might correlate with more robust internal representations that can lead to better generalization. Unfortunately, reasoning about what it means for the rank to be too low is hard in general, as the rank is agnostic to which direction of variations in the data are being ignored or to higher-order terms that hint towards a more compact representation of the data with fewer dimensions.

Our results indicate that an agent's effective rank and performance correlate in restricted settings, such as ReLU activation functions, Q-learning, and a fixed architecture. However, as we showed in our experiments, this correlation is primarily spurious in other settings since it disappears with simple modifications such as changing the activation function and the learning rate. We found several ways to improve the effective rank of the agent without improving the performance, such as using tanh instead of ReLU, an auxiliary loss (e.g., CURL), and the optimizer. These methods address the rank collapse but not the underlying learning deficiency that causes the collapse and the poor performance. Nevertheless, our results show that the dynamics of the rank and agent performance through learning are still poorly understood; we need more theoretical investigation to understand the relationship between those two factors. We also observed in Tandem and offline RL settings that the rank collapses to a minimal value early in training. Then there is unexplained variance between agents in the later stages of learning. Overall, the cause and role of this early rank collapse remain unknown, and we believe understanding its potential effects is essential in understanding large-scale agents' practical learning dynamics. The existence of low-rank but high-performing policies suggest that our networks can be over-parameterized for the tasks and parsimonious representations emerge naturally with TD-learning-based bootstrapping losses and ReLU networks in the offline RL setting. Discarding the dead ReLU units might achieve a more efficient inference. We believe this finding can give inspiration to a new family of pruning algorithms.

**Acknowledgements:**   We would like to thank Clare Lyle, Will Dabney, Aviral Kumar, Rishabh Agarwal, Tom le Paine, Mark Rowland and Yutian Chen for the discussions. We want to thank Mark Rowland, Rishabh Agarwal and Aviral Kumar for the feedback on the early draft version of the paper. We thank Sergio Gomez and Bjorn Winckler for their help with the infrastructure and the code-base at the inception of this project. We would like to thank the developers of Acme (Hoffman et al., 2020), Jax (Bradbury et al., 2018) and the Deepmind JAX ecosystem (Babuschkin et al., 2020) for developing the software infrastructure that enabled our experiments.

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

# A Appendix

## A.1 The impact of depth

Pennington and Worah (2017) explored the relationship between the rank and the depth of a neural network at initialization and found that the rank of each layer's feature matrix decreases proportionally to the index of a layer in a deep network. Here, we test the performance of a feedforward network on the bsuite task trained using Double Q-learning (Van Hasselt et al., 2016) with 2, 8, and 16 layers to see the effect of the number of layers on the effective rank. All our networks use ReLU activation functions and use He initialization (He et al., 2015). Figure 23 illustrates that the rank collapses as one progresses from lower layers to higher layers proportionally at the end of training as well. The network's effective rank (rank of the penultimate layer) drops to a minimal value on all three tasks regardless of the network's number of layers. The last layer of a network will act as a bottleneck; thus, a collapse of the effective rank would reduce the expressivity. Nevertheless, a deeper network with the same rank as a shallower one can learn to represent a larger class of functions (be less under-parametrized).

The agents exhibit poor performance when the effective rank collapses to 1. At that point, all the ReLU units die or become zero irrespective of input. Thus on bsuite, deeper networks –four and eight layered networks– performed worse than two layered MLP.

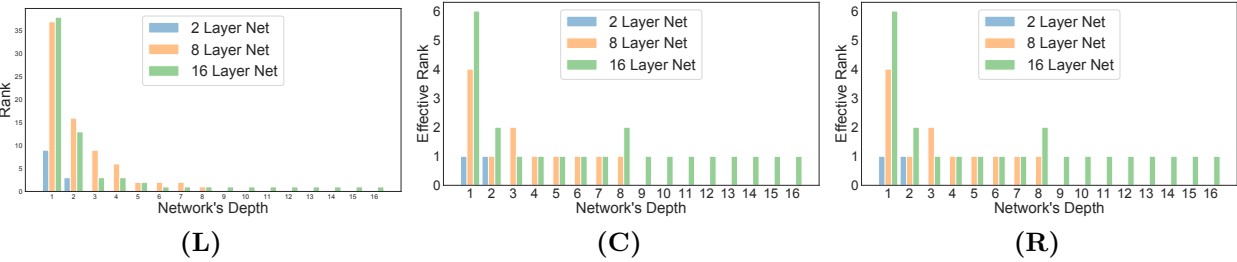

Figure 23: [**bsuite**] **The ranks and depths of the networks:** The evolution of the ranks across different layers of deep neural networks. The figure on the left **(L)** is for the ranks of catch across different layers. The figure at the center **(C)** is for the ranks of mountain_car across different layers. The figure on the right **(R)** is for cartpole.

## A.2  Spectral density of hessian for offline RL

Analyzing the eigenvalue spectrum of the Hessian is a common way to investigate the learning dynamics and the loss surface of deep learning methods (Ghorbani et al., 2019; Dauphin et al., 2014). Understanding Hessian's loss landscape and eigenvalue spectrum can help us design better optimization algorithms. Here, we analyze the eigenvalue spectrum of a single hidden layer feedforward network trained on the bsuite Catch dataset from RL Unplugged (Gulcehre et al., 2020) to understand the loss-landscape of a network with a low effective rank compared to a model with higher rank at the end of the training. As established in Figure 24, ELU activation function network has a significantly higher effective rank than the ReLU network. By comparing those two networks, we also look into the differences in the eigenvalue spectrum of a network with high and low rank. Since the network and the inputs are relatively low-dimensional, we computed the full Hessian over the dataset rather than a low-rank approximation. The eigenvalue spectrum of the Hessian with ReLU and the ELU activation functions is shown in Figure 24. The rank collapse is faster for ReLU than ELU. After 900k gradient updates, the ReLU network concentrates 92% of the eigenvalues of Hessian around zero; this is due to the dead units in ReLU network (Glorot et al., 2011). On the other hand, the ELU network has a few very large eigenvalues after the same number of gradient updates. In Figure 25, we summarize the distribution of the eigenvalues of the Hessian matrices of the ELU and ReLU networks. As a result, the Hessian and the feature matrix of the penultimate layer of the network will be both low-rank. Moreover, in this case, the landscape of the ReLU network might be flatter than the ELU beyond the notion of wide basins (Sagun et al., 2016). This might mean that the ReLU network finds a simpler solution. Thus, the flatter landscape is confirmed by the simpler function learned and less capacity used at the end of the training, which is induced by the lower rank representations.

## A.3  DeepMind lab: performance vs the effective ranks

In Figure 26, we show the correlation between the effective rank and the performance of R2D2, R2D2 trained with CURL and R2D2 trained with SAM. When we look at each variant separately or as an aggregate, we don't see a strong correlation between the performance and the rank of an agent.

## A.4  SAM optimizer: does smoother loss landscape affect the behavior of the rank collapse?

We study the relationship between smoother loss landscapes and rank collapse. We use Sharpness Aware Minimization (SAM) (Foret et al., 2021) loss for potentially creating flatter loss surfaces in order to see if smoother loss landscapes affect the rank and performance dynamics differently. Figures 27 and 28 show the evolution of feature ranks and generalization performance in Atari and DeepMind lab respectively. We do not observe a very clear relation between the extent of smoothing the loss and the feature ranks or generalization performance.

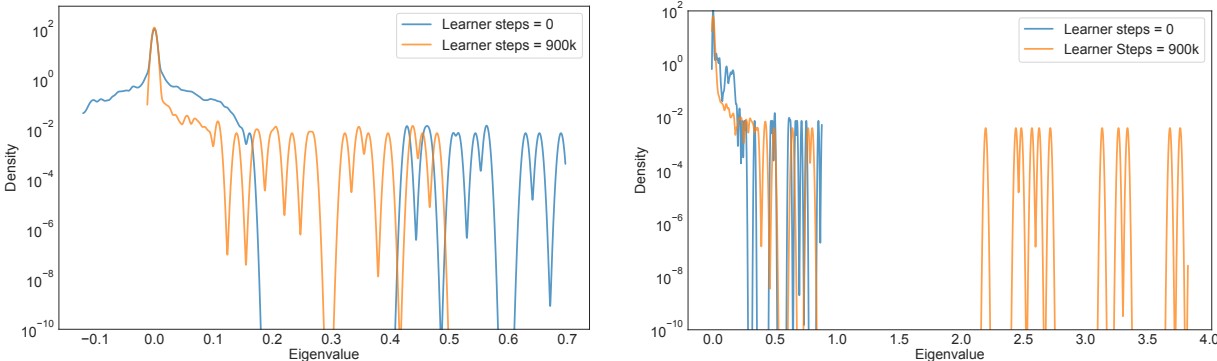

Figure 24: [**bsuite Catch**] **spectral density of Hessian:** The visualization of the spectral density of the full Hessian of a network trained with 64 units using ReLU (left) and ELU (right) activation functions. The eigenvalues of the Hessian of the offline DQN with ReLU activation function are concentrated around 0 and most of them are less than 1. The eigenvalues of the ELU network also concentrate around 0 with a few large outlier eigenvalues.

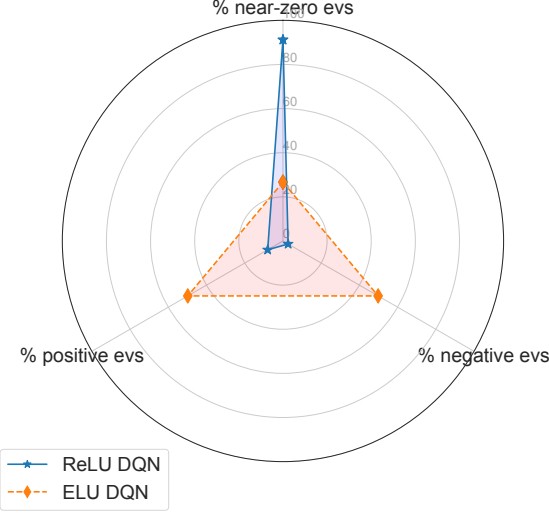

Figure 25: [**bsuite Catch**] **Hessian eigenvalues (evs) for offline DQN:** We visualize the percentages of positive, negative, and near-zero eigenvalues of the Hessian for offline DQN with ELU and ReLU activation functions on the catch dataset from bsuite. If the absolute value of an eigenvalue is less than 1e-7, we consider it as a near-zero eigenvalue. We can see that for ELU network, near-zero, positive and negative eigenvalues are almost evenly distributed. However, with ReLU network majority of eigenvalues are near-zero (90% of the evs are exactly zero), very few negative (2 %) and some positive eigenvalues (7.1 %).

## A.5    Effective rank and the value error

A curious relationship is between the effective rank and the value error, because a potential for the rank collapse or Phase 3 with the TD-learning algorithms can be the errors propagating thorough the bootstrapped targets. Figure 29 shows the correlation between the value error and the effective ranks. There is a strong anti-correlation between the effective ranks and the performance of the agents except on the levels where the expert agent that generated the dataset performs poorly at the end of the training (e.g. IceHockey.) This makes the hypothesis that the extrapolation error can cause rank collapse (or push the agent to Phase 3) more plausible.

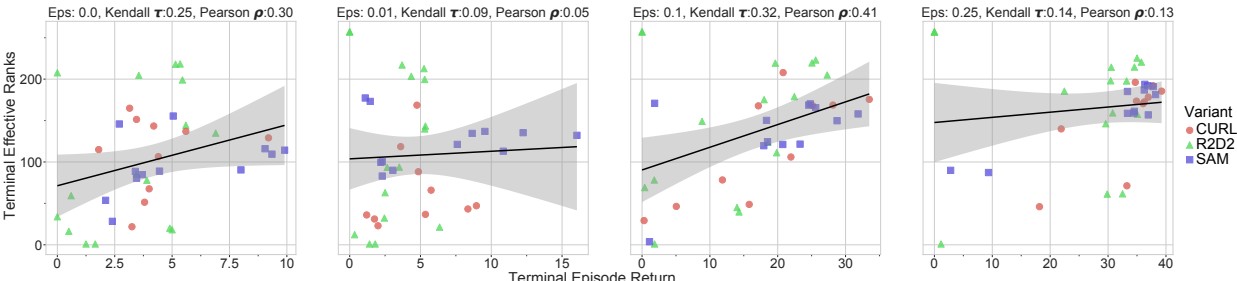

Figure 26: **[DeepMind lab] The effective rank and the performance:** Correlations between feature ranks and episode returns for different exploration noises on DeepMind lab dataset. We include data from three models: R2D2, R2D2 trained with CURL, and R2D2 trained with SAM. We do not observe a strong correlation between the effective rank and performance across different noise exploration levels in the data.

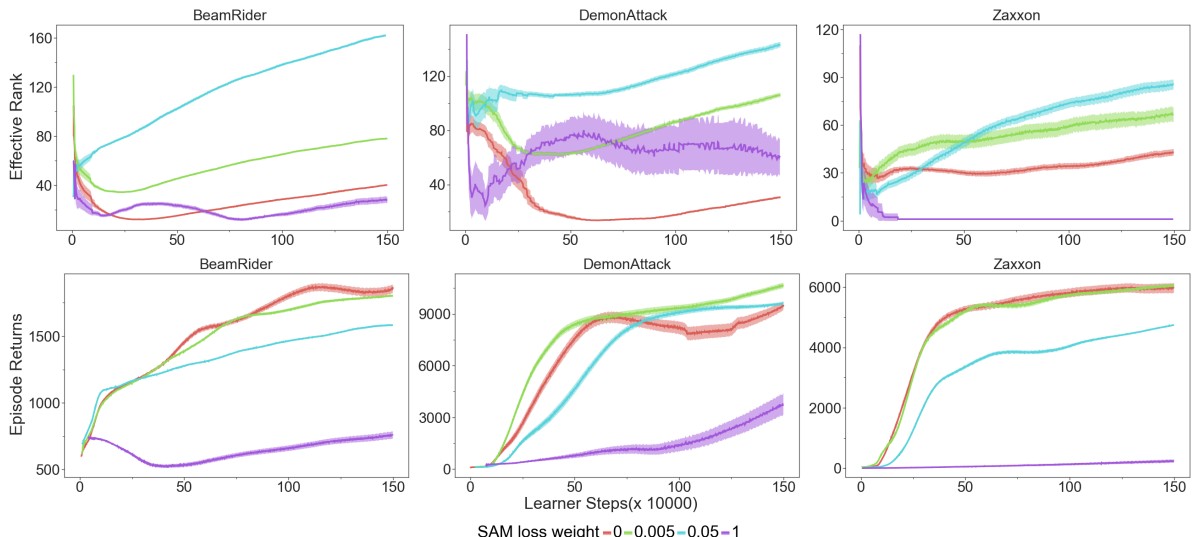

Figure 27: **[Atari] Sharpness Aware Minimization (SAM) Loss:** Evolution of ranks as we increase the weight of auxiliary losses. We see that some amount of weight on the SAM loss helps mitigate the extent of rank collapse but we observe no clear relationship with the agent performance.

## A.6 CURL on DeepMind Lab

In Figure 30 shows the CURL results on DeepMind lab dataset. We couldn't find any consistent pattern across various CURL loss weights.

## A.7 Learning rate evolution

In Figure 31, we also perform a hyperparameter selection by evaluating the model in the environment at various stages during the training. As the offline DQN is trained longer, the optimal learning rate for the best agent performance when evaluated online goes down. As one increases the number of training steps of an agent, we need to change the learning rate accordingly since the number of training steps affects the best learning rate.

## A.8 Learning curves long training regime and the different phases of learning

In this subsection, we investigate the effect of changing learning rates on the effective rank and the performance of the agent on RL Unplugged Atari online policy selection games. We train the offline DQN for

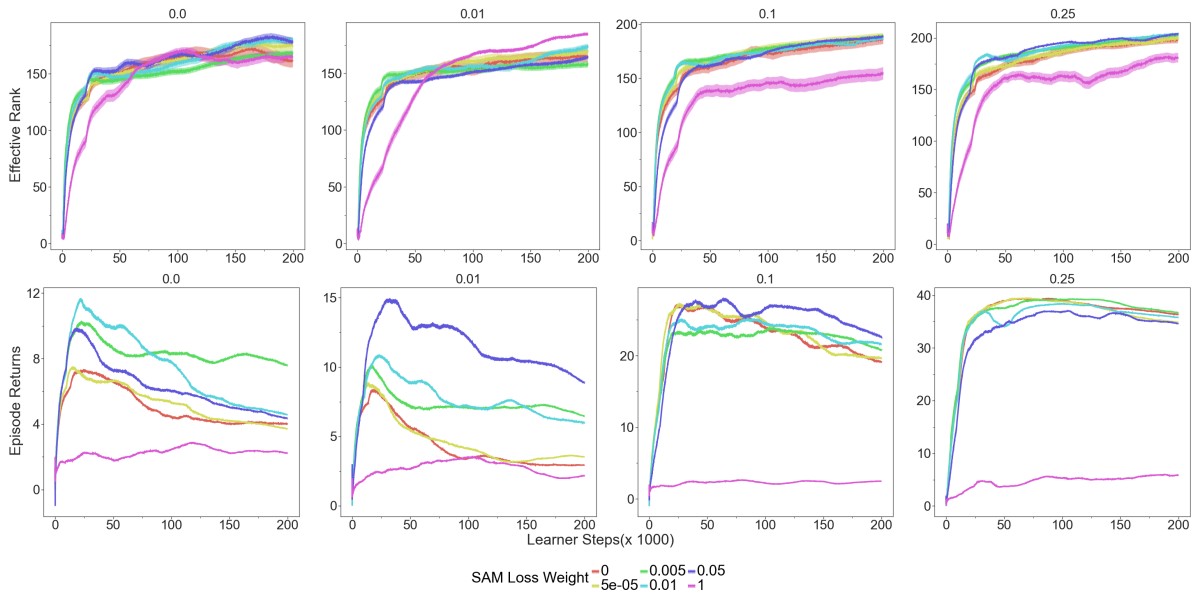

Figure 28: [**DeepMind lab-SeekAvoid Snapshot**] **Sharpness Aware Minimization (SAM) Loss** We do not observe any rank collapse as we continue training with the LSTM networks (because of the tanh activations discussed in Section 6) used in DeepMind lab dataset over a spectrum of different weights for the SAM loss.

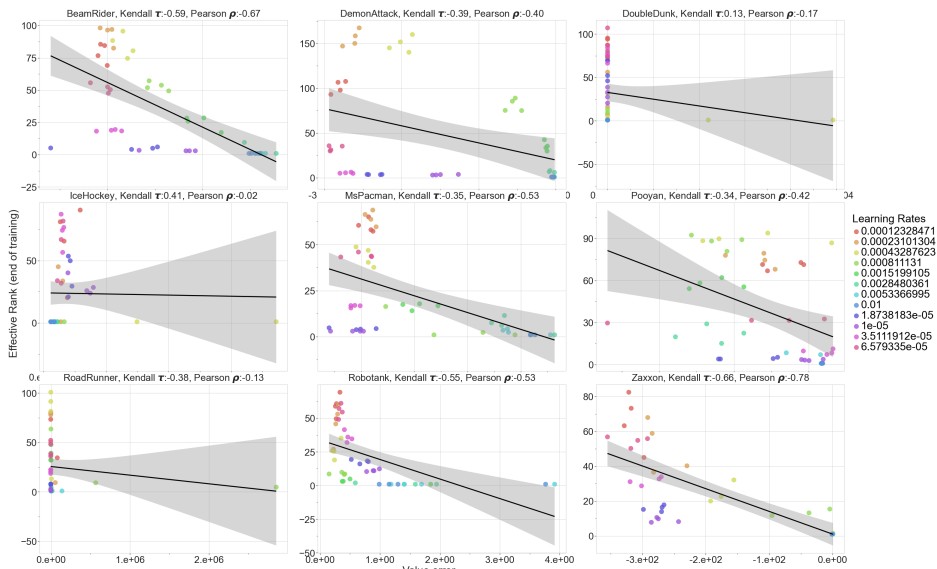

Figure 29: [**Atari**]: These plots shows the correlation between the value error and the effective rank for offline DQN agent trained for 20M steps on online policy selection games for Atari. There is an apparent anti-correlation between the effective rank and the value error. Namely, as the value error of an agent when evaluated in the environment increases the effective rank decreases. The correlation is significant on most Atari levels except IceHockey where the expert agent that generated the dataset performs poorly.

20M learning steps which is ten times longer than typical offline Atari agents (Gulcehre et al., 2020). We evaluated twelve learning rates in $[10^{-2}, 10^{-5}]$ equally spaced in logarithmic space. We show the effective rank learning and performance curves in Figure 32. It is easy to identify different phases of learning in most of those curves.

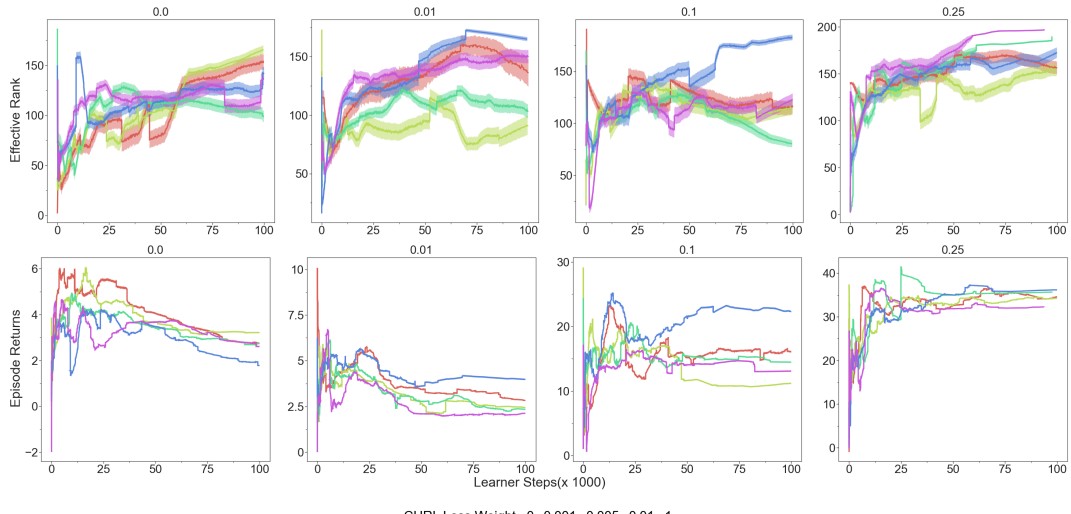

Figure 30: [**DeepMind lab-SeekAvoid:**] **auxiliary losses** Evolution of ranks as we increase the weight of auxiliary loss. While some auxiliary loss helps the model perform well, there is no clear correlation between rank and performance.

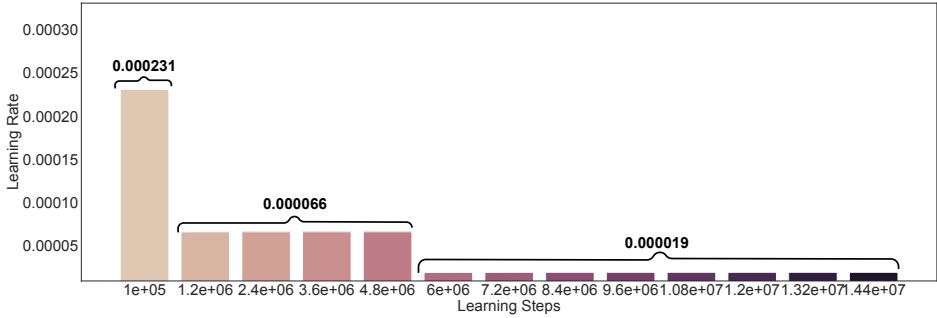

Figure 31: [**Atari**]: Evolution of the optimal learning rate found by online evaluations in the environment. As the model is trained longer the optimal learning rate found by online evaluations goes down.

### A.9 Effective rank and the performance

Figure 33 shows the correlation between the effective rank and the performance of an agent trained with minibatch of size 32. On most Atari online policy selection games it is possible to see a very strong correlation but on some games the correlation is not there. It seems like even

Figure 34 depicts the correlation between the effective rank and the performance of a DQN agent with ReLU activation function. There is a significant correlation on most Atari games. As we discussed earlier, long training setting with ReLU activation functions where the effect of the rank is the strongest on the performance.

### A.10 Computation of feature ranks

Here, we present the *Python* code-stub that we used across our experiments (similar to Kumar et al. (2020a)) to compute the feature ranks of the pre-output layer's features:

```python
import numpy as np
def compute_rank_from_features(feature_matrix, rank_delta=0.01):
  """Computes rank of the features based on how many singular values are significant."""
```

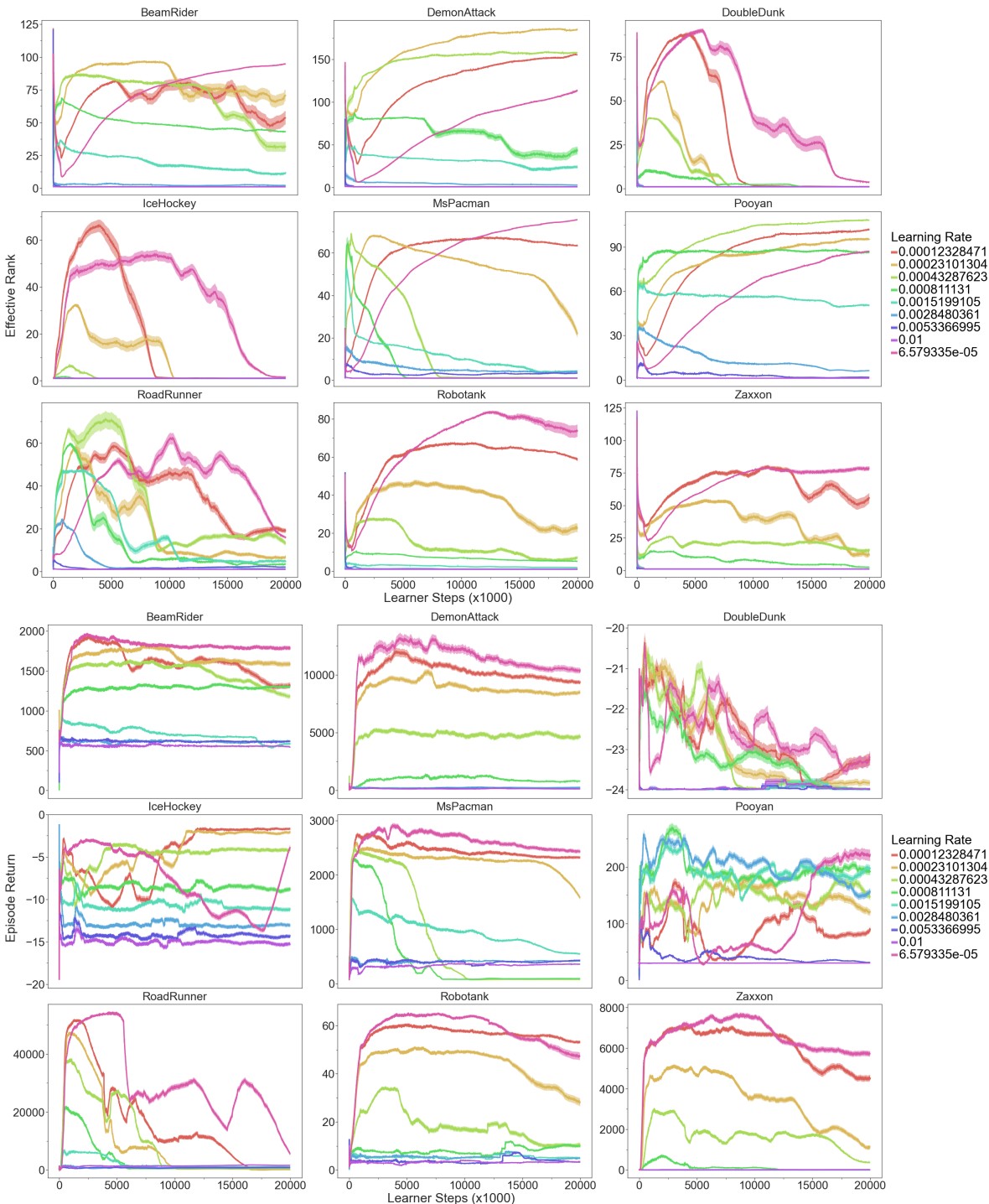

Figure 32: [**Atari**] **Effective rank curves:** Effective rank and performance offline DQN agent across nine Atari online policy selection games. We train the offline DQN agent for 20M learning steps and evaluated 12 learning rates in $[10^{-2}, 10^{-5}]$ equally spaced in logarithmic space. Let us note that in all games rank goes down early in the training (Phase 1), then goes up (phase 2) and for some learning rate the effective rank collapses (Phase 3). Correspondingly, the performance is low in the beginning of the training (Phase 1), goes up and stays high for a while (Phase 2), and sometimes it the performance collapses (Phase 3).

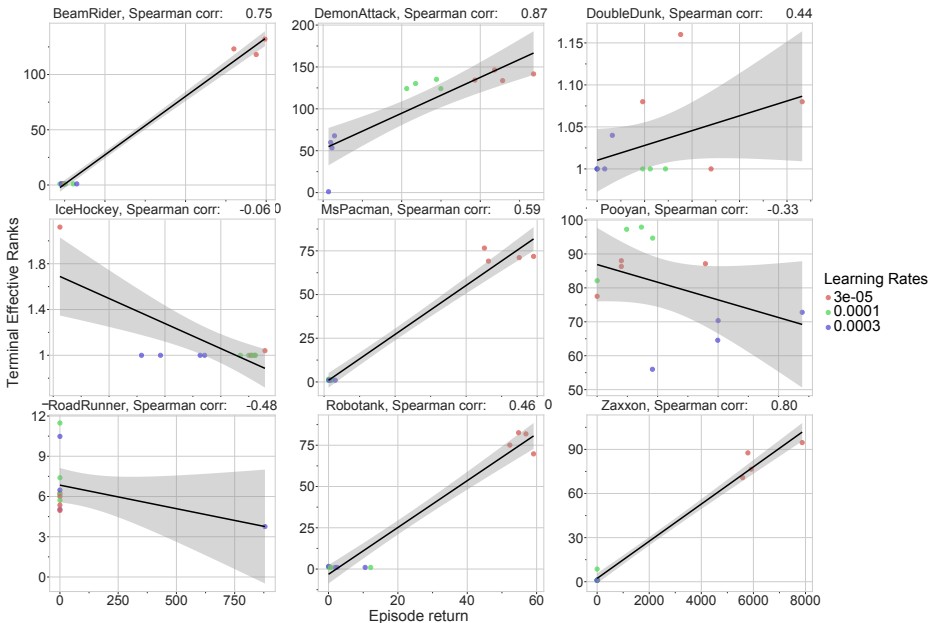

Figure 33: [**Atari**] The correlation between ranks and returns for DQN trained with minibatch size of 32. We ran each network with three learning rates and five different seeds but we only show the mean across those five seeds here. We can see very strong correlations on some games, but that correlation is not consistent.

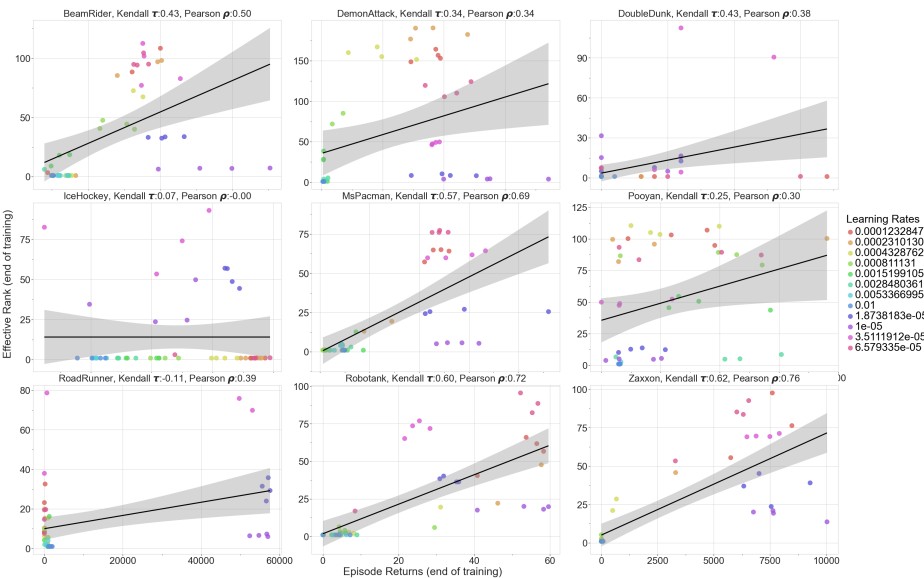

Figure 34: [**Atari**] The correlation between ranks and returns for DQN trained for a network trained for 20M learning steps and 12 different learning rates with minibatch size of 256. We can see that there is significant positive correlation between the rank and the learning rates for most online policy selection games.

| Hyper-parameters | bsuite | Atari | DeepMind lab |
|---|---|---|---|
| Training batch size | 32 | 256 | 4 (episodes) |
| Rank calculation batch size | 512 | 512 | 512 |
| Num training steps | 1e6 | 2e6 | 2e4 |
| Learning rate | 3e-4 | 3e-5 | 1e-3 |
| Optimizer | Adam | Adam | Adam |
| Feedforward hidden layer size | 64 | 512 | 256 |
| Num hidden layers | 2 | 1 | 1 |
| Activation | ReLU | ReLU | ReLU (tanh for LSTM gates) |
| Memory | None | None | LSTM |
| Discount | 0.99 | 0.99 | 0.997 |

Table 3: The default hyper-parameters used in our work across different domains.

```
sing_values = np.linalg.svd(feature_matrix, compute_uv=False)
cumsum = np.cumsum(sing_values)
nuclear_norm = np.sum(sing_values)
approximate_rank_threshold = 1.0 - rank_delta
threshold_crossed = (
    cumsum >= approximate_rank_threshold * nuclear_norm)
effective_rank = sing_values.shape[0] - np.sum(threshold_crossed) + 1
return effective_rank
```

### A.11   Hyperparameters

Here, we list the standard set of hyper-parameters that were used in different domains: bsuite, Atari, and DeepMind Lab respectively. These are the default hyper-parameters, which may differ when stated so in our specific ablation studies. For the DMLLAB task, we use the same network that was used by Gulcehre et al. (2021). For all the Atari tasks, we use the same convolution torso that was used by Gulcehre et al. (2020) which involves three layers of convolution with ReLU activations in between.

- Layer 1 - Conv2d(channels=32, kernel_shape=[8, 8], stride=[4, 4])

- Layer 2 - Conv2d(channels=64, kernel_shape=[4, 4], stride=[2, 2])

- Layer 3 - Conv2d(channels=64, kernel_shape=[3, 3], stride=[1, 1])

### A.12   bsuite phase transitions and bottleneck capacity

We illustrate the phase transition of a simple MLPs with ReLU activations. In Figure 35, we have a network of size $(64, \text{bottleneck units}, 64)$ where we vary the number of bottleneck units. In Figure 36, we have a network of size $(64, \text{bottleneck units})$ where we vary the number of bottleneck units. In both cases, having smaller number of bottleneck units reduces the performance of the mode and agents were able to solve the problem even when the penultimate layer's effective rank was small. With the larger learning rate the right handside figures (b), the effective ranks tend to be lower.

### A.13   Activation sparsity on bsuite

In Figure 37, we show that the activations of the ReLU network becomes very sparse during the course of training. The sparsity of the ReLU units seems to be significantly higher than the ELU units at the end of training.

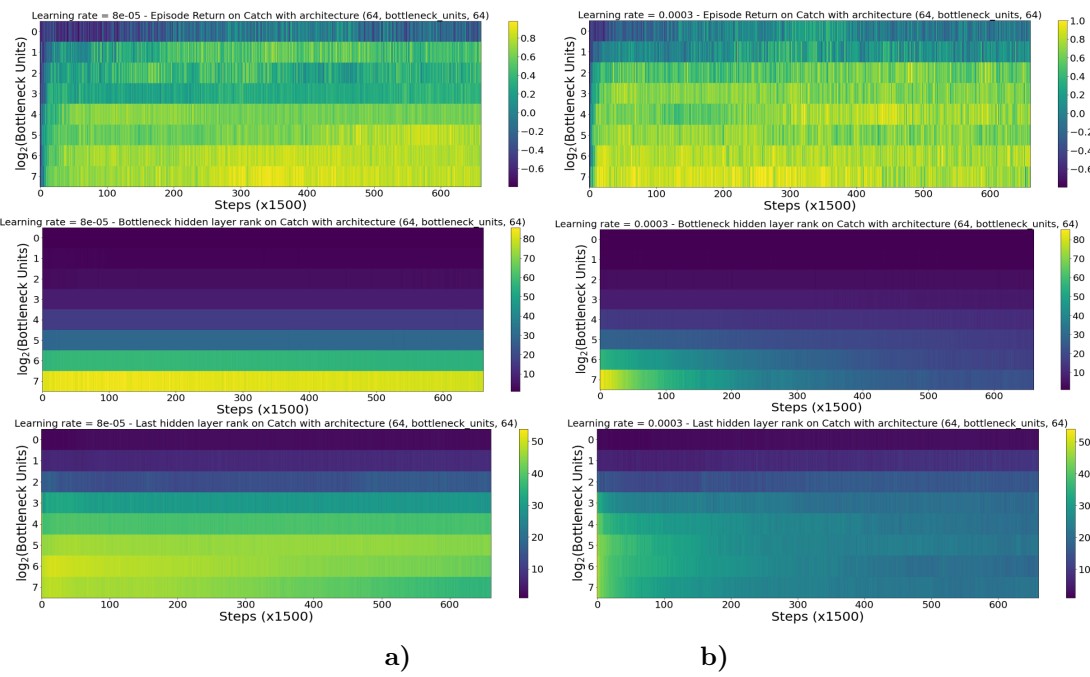

**a)**        **b)**

Figure 35: [**bsuite**] **- Catch dataset:** Phase transition plots of a network with hidden layers of size $(64, \text{bottleneck units}, 64)$. On the y-axis, we show the $\log_2(\text{bottleneck units})$, and x-axis is the number of gradient steps. The figures on the left hand-side (a) is trained with the learning rate with 8e-5 and right hand-side (b) experiments are trained with learning rate of 3e-4. The first row shows the episodic returns. The second row shows the effective rank of the second layer (bottleneck units) and the third row is showing the penultimate layer's effective rank. The effective rank collapses much faster with the learning rate of 4e-3 than 5e-8. The low ranks can still have good performance. The low bottleneck units causes the effective rank of the last layer to collapse faster. The performance of the network with the small number of bottleneck units is poor. The effective rank of the small number of bottleneck units is smaller.

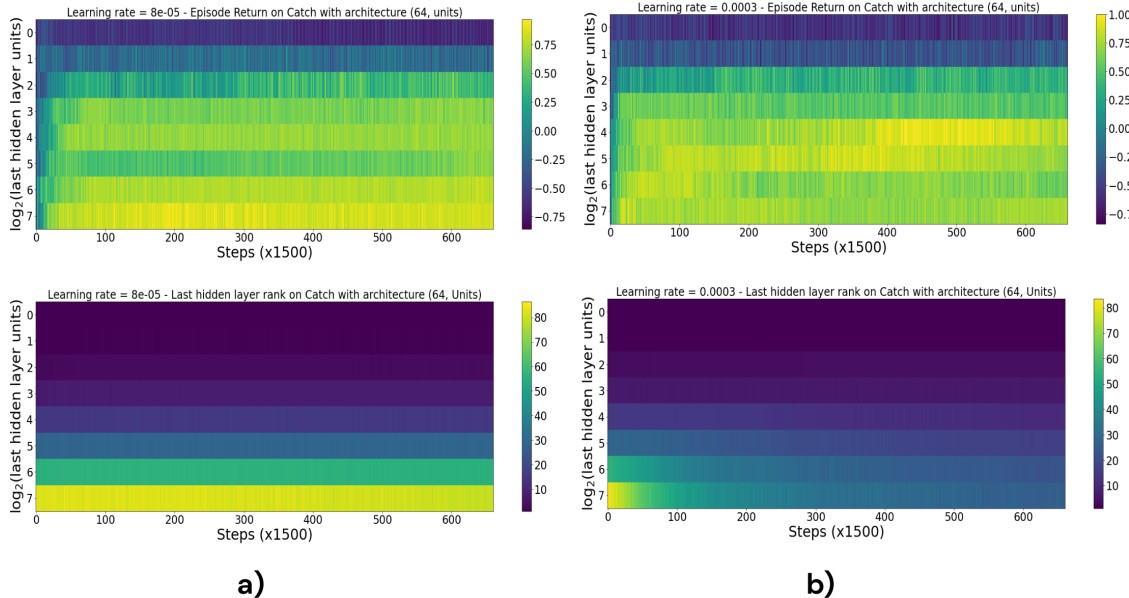

Figure 36: [**bsuite**] **- Catch dataset:** Phase transition plots of a network with hidden layers of size $(64, \text{bottleneck units})$. On the y-axis, we show the $\log_2(\text{bottleneck units})$, and x-axis is the number of gradient steps. The figures on the left hand-side (a) is trained with the learning rate with 8e-5 and right hand-side (b) experiments are trained with learning rate of 3e-4. The first row shows the episodic returns. The second row shows the effective rank of the second layer (bottleneck units). The effective rank collapses much faster with the learning rate of 4e-3 than 5e-8. The low ranks can still have good performance. The small number of bottleneck units causes the effective rank of the layer to collapse faster. The performance of the network with the small number of bottleneck units is poor. The effective rank of the small number of bottleneck units is smaller.

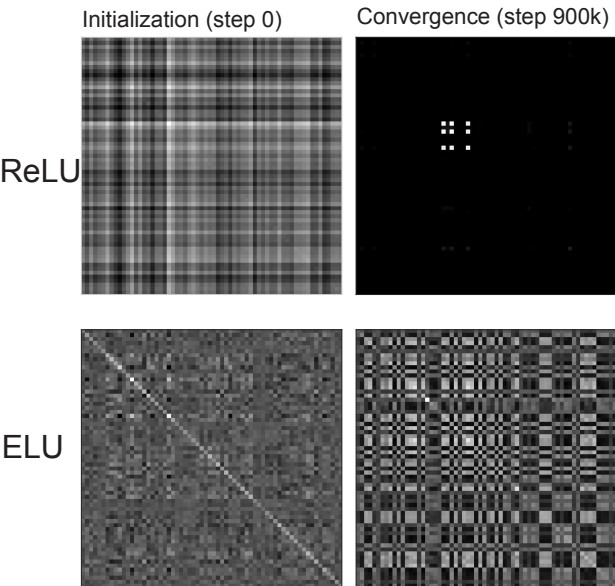

Figure 37: [**bsuite Catch**] **Gram matrices of activations:** Gram matrices of activations of a two-layer MLP with ReLU and ELU activation functions. The activations of the ReLU units become sparser when compared to ELU units at the end of the training due to dead ReLU units.

