# OpenReview forum: "An empirical study of implicit regularization in deep offline RL"
_TMLR — Accepted by TMLR_

### Review · Reviewer_5mXC · 2022-09-10

**Summary Of Contributions:**

This paper investigates the hypothesis that (effective) rank collapse causes poor performance in offline RL. Through an extensive set of experiments on offline RL benchmarks, the authors show that the results identified in past work only hold for a small range of hyperparameters, and so targeting it may not be a viable solution to improving performance. They also identify several interesting phases of offline RL training related to the rank and performance.

**Broader Impact Concerns:**

None.

**Requested Changes:**

Changes that are critical for me to recommend acceptance:
- Discussion: as discussed in the weaknesses, I believe the observation by Kumar et al. (2020) that controlling rank could improve performance should be better-discussed. Is this also a question of hyperparameter range, or do you believe there are confounding factors?

Changes that would strengthen paper:
- Figure 2: it’d be useful for the reader to know if the y-axes here start at zero (rank / returns) or not.
- Section 2.2: I’m confused by the usage of the word “ablate” here. Usually it means to remove certain components to check their effects, but here it seems to just mean you use them? Perhaps writing here could be made more clear.

**Strengths And Weaknesses:**

Below are my views on the strengths and weaknesses of the results. Note that I am not very familiar with RL and so am not very qualified to comment on the appropriateness or quality of the experiments.

Strengths:
- The paper identifies an interesting hypothesis and does a very thorough investigation of it.
- The authors identify numerous hyperparameters that may affect the outcome and study their ablations.
- The writing is fairly straightforward to understand.
- In addition to the main objective of the work, there is also an interesting discussion and empirical investigation of training dynamics in RL.

Weaknesses:
- One important discussion that seems missing to me is an explanation for why Kumar et al. (2020) observed improved performance when controlling rank collapse.
- W.r.t. the definition of effective rank: I understand that the authors are aiming to align with Kumar et al. (2020), but since this “effective rank” quantity is effectively an attempt to discount small singular values, I wonder if a better quantity to study—since it is parameter-free—would be the stable rank (c.f. the actual quantity used in Arora et al. (2018) and Sanyal et al. (2020)). Of course, these papers would argue instead for a smaller rather than a larger (stable) rank.

---

> ### Author Response · Authors · 2022-09-16
> **Response to Review of Paper374 by Reviewer 5mXC**
>
> We would like to thank the reviewer for their valuable comments. We reply to their comments below:
>
> > w.r.t. the definition of effective rank: I understand that the authors are aiming to align with Kumar et al. (2020), but since this “effective rank” quantity is effectively an attempt to discount small singular values, I wonder if a better quantity to study—since it is parameter-free—would be the stable rank (c.f. the actual quantity used in Arora et al. (2018) and Sanyal et al. (2020)). Of course, these papers would argue instead for a smaller rather than a larger (stable) rank.
>
> To be consistent with Kumar et al. 2020, we used their paper's definition of effective rank. However, since it is parameter-free, we considered using the effective rank as defined in [1] which is used by Arora et al. (2018.) When we compared the "parameter-free effective rank" (by [1]) with the matrix rank (as implemented in the numpy linalg library) and the effective rank from Kumar et al. 2020 (for $\delta = 0.01$) in our preliminary experiments, we found that all those measures almost perfectly correlate with each other. As a result, we decided to use their definition of effective rank to be consistent with Kumar et al. 2020.
>
> Thanks a lot for the other rank definitions that you pointed out. In the camera-ready version of the paper, we will elaborate on the differences and similarities between all these metrics and elaborate on ways of measuring the effective rank in a separate section.
>
> > One important discussion that seems missing to me is an explanation for why Kumar et al. (2020) observed improved performance when controlling rank collapse.
> > Discussion: as discussed in the weaknesses, I believe the observation by Kumar et al. (2020) that controlling rank could improve performance should be better-discussed. Is this also a question of hyperparameter range, or do you believe there are confounding factors?
>
> We discussed with the authors of (Kumar et al. 2020) a year ago, sharing some of our early results, and they added the following observation in their latest manuscript on arxiv:
>
> >... while our proposed penalty is effective in many cases in offline and online settings, it does not solve the problem fully, i.e., it does not address the root cause of implicit under-parameterization and only addresses a symptom, and a more sophisticated solution may better prevent the issues with implicit under-parameterization
>
> Kumar et al., 2020 is not claiming a strong causal relationship between rank and performance. However, several parts of the paper mention that a  relationship between effective rank and performance exists, which in our paper, we showed is only true in restricted settings. In our discussions with the authors, we concluded that they observe the positive association between rank and performance because the model is getting into Phase 3 (Under-parameterization) due to large learning rates and small (32 instead of 256) mini-batches used in the paper. We claim that phase 3 can be detrimental to the performance of offline RL algorithms. Having said that, it is relatively easy to avoid it as prescribed in our paper, for example, by lowering the learning rate or changing the activation function. We will make this more explicit in the paper. In Figure 1, the line with a blue shade is a model using hyperparameters sweep similar to the one used by Kumar et al., 2020. The only difference is that we used Adam instead of RMSProp, and in that setting, it is possible to see the positive association between rank and performance on most Atari games. We will add a further discussion about this comparison to the paper.
>
>
> > Figure 2: it’d be useful for the reader to know if the y-axes here start at zero (rank / returns) or not.
>
> Thanks, that is a good point. We will update this figure to make it more clear.
>
> > Section 2.2: I’m confused by the usage of the word “ablate” here. Usually it means to remove certain components to check their effects, but here it seems to just mean you use them? Perhaps writing here could be made more clear.
>
> Thanks for pointing this out. We will improve the writing in that section. We agree that the “ablate” is not appropriate here, we will rephrase it to “..., we test the effect of using an …”.
>
> ---
>
> We will make the updates suggested by the reviewer once the other reviews are available and update the paper with a summary of the changes made to it.
>
>
> [1] Roy, O. and Vetterli, M., 2007, September. The effective rank: A measure of effective dimensionality. In 2007 15th European signal processing conference (pp. 606-610). IEEE.

---

### Review · Reviewer_hEii · 2022-09-26

**Summary Of Contributions:**

This submission provides an extensive empirical investigation on the relationship between effective rank and agents' performance in deep offline reinforcement learning. The authors proposed a few hypotheses on connecting ranks with different ingredient in deep neural networks. Experiments are provided to validate and discuss the hypotheses.

**Requested Changes:**

It would be more convincing if the authors can provide either (1) theoretical justification on the relationship between rank and performance (2) more convincing empirical justification to draw some "specific" conclusion. A few detailed questions and comments are provided as follows.

1. In the caption for Figure 5, the authors stated that "the ranks are distributed almost like a Gaussian". More rigor is necessary, instead of simple guesses.

2. The authors mentioned in the caption of Figure 16 that "auxiliary loss helps mitigate the rank collapse but prevents the model from learning useful representations". Based on my understanding, there are no steps to measure the usefulness of representations and hence such a measure is necessary before drawing this conclusion.


**Strengths And Weaknesses:**

Strengths
1. Investigation the relationship between rank of the penultimate layer of neural networks and agents' performance is an interesting topic, and may motivate more follow-up work.

2. The authors did provide an extensive investigation on the topic from different angles, which is a good first attempt to tackle this problem.


Weakness
1. The current version is over empirical and hence it's difficult to draw any concrete conclusion. For example, on Key Observation 2 in Page 3, it's extremely difficult to verify that the three stages are necessary. Furthermore, the definitions for "simple" and "complex" are lacking and hence we cannot distinguish them with clear boundaries.

2. As mentioned above, this submission cannot draw any conclusion based on the results shown. It's true that there is somewhat correlation between the effective rank and performance. However, we cannot know when it will or not will be the case from the provided results.

---

> ### Author Response · Authors · 2022-09-30
> **In response to lack of theoretical results and weaknesses**
>
> Thanks a lot for taking the time to review our paper and opening up good discussion points.
>
> > The current version is over empirical and hence it's difficult to draw any concrete conclusion … (1) theoretical justification on the relationship between rank and performance
>
> We want to position our work as a comprehensive and empirical study of the relationship between effective rank and performance. Previous works, such as Kumar et al. 2020 [1], presented some theories on rank collapse with TD-Learning. However, as we argue in the paper, such theories are limited because they are based on a simplistic and largely impractical assumption of deep linear neural networks and infinitesimal learning rates (see Section 4.2 of Kumar et al. 2020.) Moreover, the theories they established do not address the relationship between rank and performance. To the best of our knowledge, there are no studies that analyze the generalization and learning dynamics of deep nonlinear networks with "rank collapse" from a theoretical perspective. Unfortunately, we do not have a robust and reliable theoretical framework for analyzing the performance and generalization of deep neural networks [3]. Thus even the papers investigating the relationship between the effective rank and performance in the supervised learning setting focus on empirical results (such as [2].) Our work provides an empirical illustration of cases under which conditions such collapse can be detrimental. *However, we agree with the reviewer that a theoretical understanding of the relationship between the effective rank and the performance would be very valuable and we hope that the empirical observations made in this work will inspire more theoretical investigation in this direction which we point out in the Discussion section.*
>
> > Weakness 1, ... for example, on Key Observation 2 in Page 3, it's extremely difficult to verify that the three stages are necessary.
>
> Regarding the comment on the three stages of learning, we wanted to emphasize that it is not straightforward to predict which phase a particular training will end up in. Generally, we observe that training a network long enough does allow it to cross phases, as shown in Figure 5 and other similar ablations. None of the phases except phase 1 is “necessary” as networks could get stuck in phase 1 under very low learning rates and short durations of training. Nevertheless, we have also observed that we did see phase 1 and phase 2 in most of our ablations with a reasonable range of learning rates both in BC agents and DQN (TD-Learning) agents. In contrast, phase 3 was unique to ReLU-network DQN agents, which happens when they are trained long enough. We wanted to demonstrate the previously published ideas of rank collapse under the lens of different phases of TD Learning. We must be careful about making sweeping generalizations based on observing the relationship between rank and performance in one phase. From our experiments, we also noticed that rank collapse did not have any consistent negative effect on the agent performance unless the rank collapses to catastrophically low numbers such as 1, from which the network does not recover.
>
> As we stated in our paper, if one can establish a causal link between rank and performance under a wide variety of settings, it could open up several avenues in fully offline RL such as model selection, early stopping etc. Having said that, our extensive ablations show that such a conclusion is not straightforward at all and we need to understand several other confounding factors are in play to determine the causal link between the two quantities. We believe that our analysis and this key observation is useful for Offline RL practitioners and at the same time, might motivate future work on deep-RL theory to understand this relationship better in the presence of these confounders.
>
>
> ### References
>
> [1]  Kumar et al. (2020), Implicit Under-Parameterization Inhibits Data-Efficient Deep Reinforcement Learning, https://arxiv.org/abs/2010.14498.
>
> [2] Huh, Minyoung, Hossein Mobahi, Richard Zhang, Brian Cheung, Pulkit Agrawal, and Phillip Isola. "The low-rank simplicity bias in deep networks." arXiv preprint arXiv:2103.10427 (2021).
>
> [3] Jiang, Y., Natekar, P., Sharma, M., Aithal, S.K., Kashyap, D., Subramanyam, N., Lassance, C., Roy, D.M., Dziugaite, G.K., Gunasekar, S. and Guyon, I., 2021, August. Methods and analysis of the first competition in predicting generalization of deep learning. In NeurIPS 2020 Competition and Demonstration Track (pp. 170-190). PMLR.

---

### Review · Reviewer_8ZFd · 2022-11-03

**Summary Of Contributions:**

The authors conduct an extensive empirical investigation into the phenomenon of implicit under parametrization in offline (deep) Q learning (introduced by Lavine et al in 2020), i.e a situation where high capacity functional aproximators trained via TD learning are unable to fit the data because the final layer of activation "collapses" to a highly restricted subspace. They conclude that while one can indeed observe this behavior in some settings, the existence of implicit underparametrization is highly sensitive to network hyperparameters such as choice of activation function, learning rate, minibatch size etc. They conclude that extrapolating from simplified models (where one can analytically prove that implicit under parametrization occurs) and specific hyperparameter settings (where one can verify its occurence empirically) is misleading, and that the precise role of implicit underparametrization in the performance of deep RL methods is still unclear.

**Broader Impact Concerns:**

I don't see any particular ethical concerns with the paper.

**Requested Changes:**

Given that the authors of the original implicit underparametrization paper proved that the phenomenon occurs in kernel regression, it would have been interesting to examine the behavior of wide (but finite) networks. I should emphasize that I don't view the inclusion of tests of this regime to be critical for securing my reccomendation for acceptance.

**Strengths And Weaknesses:**

I found the presentation to be clear and engaging. In particular the authors were careful to distinguish between correlation and causation- clearly the latter is what we are interested in here, and so I appreciated that an explicit causal graph was hypothesized and tested.

The authors seem to have done a good job of conducting a comprehensive set of empirical tests, and make a convincing case that it would be misleading to claim that implicit underparametrization/rank collapse provides a monocausal explanation of the failures of deep Q-learning. I should say however that I am far from an expert in this field, and don't consider myself well qualified to judge the strength of the empirical evidence marshalled by the authors.

The authors suggest a three phase picture of Q learning- a first phase where "simple" behaviors are learned, a second phase where "complex" behaviors are learned, and a final phase of underparametrization. Given the subjective terminology, was unclear to me how exactly the authors defined these phases, and would have appreciated more details.

---

> ### Author Response · Authors · 2022-11-13
> **On simple and complex behaviors and Width Experiments.**
>
>
> We would like to thank the reviewer for their insightful feedback and comments.
>
> > The authors suggest a three phase picture of Q learning- a first phase where "simple" behaviors are learned, a second phase where "complex" behaviors are learned, and a final phase of underparametrization.
>
> Thanks for pointing out this. By *simple behaviors*, we mean the type of behaviors that emerge early during the training of an offline RL algorithm. The simple behaviors give rise to low rewards, meaning that the policy's performance would be equal or marginally better than the random policy when it is evaluated in the environment in an online fashion.d *Complex behaviors* are often learned later in training. They would achieve high rewards, usually close to the best policy in the dataset and significantly better than the random policy when the policy learned these complex behaviors are evaluated online. We will clarify this in our paper.
>
> > Given that the authors of the original implicit underparametrization paper proved that the phenomenon occurs in kernel regression, it would have been interesting to examine the behavior of wide (but finite) networks. I should emphasize that I don't view the inclusion of tests of this regime to be critical for securing my reccomendation for acceptance.
>
> Thanks for this observation. Having more ablations towards understanding the phenomenon in wide networks would be insightful. On this point, we have some preliminary experiments in **Appendix A.13** "*bsuite phase transitions and bottleneck capacity*". In our phase transition plots, we observed that making the penultimate layer or the intermediate layer wider makes the collapse slower. Still, eventually, the effective rank of the wider network collapses. However, in our final manuscript, we will note including a larger-scale experiment as a future research direction. Our current work is essentially a practical and empirical study to identify whether the relationship between effective rank and performance is ubiquitous in all settings of TD Learning and whether the proposed causal relationship is affected by confounding variables. Therefore, we did not delve much into the form of formal analysis. A comprehensive formal analysis of our different observations, such as the causal link between under-parameterization and performance, the role of confounders, and the three phases of learning, particularly in non-linear networks, would be a valuable research contribution to succeeding works.

---

### Decision · Action_Editors · 2022-12-14

**Recommendation:** Accept as is

**Comment:**

Reviewer hEii raised some concerns about the submission being overly empirical and lacking theoretical grounding.  While I agree that it would have been much stronger had it included backing theory, I think the empirical contribution is of sufficient novelty and significance for publication, and the text is completely transparent about its empirical nature.  I therefore recommend acceptance.

**Audience:**

The submission combines the timely topics of offline reinforcement learning and implicit regularization.  As such, I believe it is of interest to both reinforcement learning and theory of machine learning audiences.

**Claims And Evidence:**

The claims made in the submission are generally well supported through experiments.  There were certain arguments (e.g. the three phases of Q learning) which the reviewers found insufficiently well defined, but the authors' responses addressed raised concerns.